# *EGFRAP* encodes a new negative regulator of the EGFR acting in both normal and oncogenic EGFR/Ras-driven tissue morphogenesis

Jennifer Soler Beatty[1], Cristina Molnar[2¤a], Carlos M. Luque[2¤b], Jose F. de Celis[2], María D. Martín-Bermudo[1]*

1 Centro Andaluz de Biología del Desarrollo, Universidad Pablo de Olavide/CSIC/JA, Sevilla, Spain,
2 Centro de Biología Molecular Severo Ochoa (UAM/CSIC), Univ. Autónoma de Madrid, Madrid, Spain

¤a Current address: Institute of Research in Biomedicine (IRB Barcelona), Barcelona, Spain
¤b Current address: Department of Genetics, University of Cambridge, Cambridge, United Kingdom
* mdmarber@upo.es

**Data Availability Statement:** All relevant data are within the manuscript and its Supporting Information files.

## Abstract

Activation of Ras signaling occurs in ~30% of human cancers. However, activated Ras alone is insufficient to produce malignancy. Thus, it is imperative to identify those genes cooperating with activated Ras in driving tumoral growth. In this work, we have identified a novel EGFR inhibitor, which we have named *EGFRAP*, for EGFR adaptor protein. Elimination of *EGFRAP* potentiates activated Ras-induced overgrowth in the *Drosophila* wing imaginal disc. We show that EGFRAP interacts physically with the phosphorylated form of EGFR via its SH2 domain. EGFRAP is expressed at high levels in regions of maximal EGFR/Ras pathway activity, such as at the presumptive wing margin. In addition, EGFRAP expression is up-regulated in conditions of oncogenic EGFR/Ras activation. Normal and oncogenic EGFR/Ras-mediated upregulation of EGRAP levels depend on the Notch pathway. We also find that elimination of *EGFRAP* does not affect overall organogenesis or viability. However, simultaneous downregulation of EGFRAP and its ortholog PVRAP results in defects associated with increased EGFR function. Based on these results, we propose that *EGFRAP* is a new negative regulator of the EGFR/Ras pathway, which, while being required redundantly for normal morphogenesis, behaves as an important modulator of EGFR/Ras-driven tissue hyperplasia. We suggest that the ability of *EGFRAP* to functionally inhibit the EGFR pathway in oncogenic cells results from the activation of a feedback loop leading to increase *EGFRAP* expression. This could act as a surveillance mechanism to prevent excessive EGFR activity and uncontrolled cell growth.

## Author summary

Activation of Ras signalling occurs in ~30% of human cancers. However, activated Ras alone is insufficient to produce malignancy. Thus, the discovery of genes cooperating with Ras in cancer is imperative to understand tumoral growth driven by Ras activating mutations. A key output of over-activated EGFR/Ras signalling is the induction of a complex

**Funding:** This work was funded by the by the Spanish Agencia Estatal de Investigación (MCUI/ AEI, http://www.ciencia.gob.es/; Grant numbers BFU2016-80797-R, PID2019-109013GB-100 and MDM-2016-0687 to MDM-B; PGC2018-094476-B-100 to JFC and BFU2015-67266-R to CML) and from the Consejería de Educación e Investigación, Comunidad de Madrid (https://www.comunidad. madrid/; Grant number S2010/BMD-2305 to CML) and by the European Regional Development Fund (http://ec.europa.eu/regional_policy/en/funding/ erdf/). Core funding to the CABD from the Junta de Andalucía is acknowledged. JSB was supported by a FPI Fellowship from MICINN. The funders had no role in study design, data collection and analysis, decision to publish, or preparation of the manuscript.

**Competing interests:** The authors have declared that no competing interests exist.

and dynamic set of transcriptional networks leading to changes in gene expression. As a result of these changes, the normal function of some genes can become adjusted in a tumorigenic context. In this work, using the *Drosophila* wing imaginal disc as model system, we have identified a new EGFR inhibitor, *EGFRAP*, which function is redundant for proper morphogenesis, yet becomes an important limiter of the overgrowth driven by oncogenic EGFR/Ras activity. We show that the specificity of *EGFRAP* in cells with high levels of EGFR activity arises from activation of a negative feedback loop resulting in increased EGFRAP levels. This could act to prevent excessive EGFR activity and uncontrolled cell growth. We believe the identification of other factors behaving like *EGFRAP*, will help in our fight against cancer, as it might lead to the identification of new therapeutic drugs affecting cancer but not normal cells, a top priority in cancer research.

## Introduction

Cancer is a devastating disease that threatens human health worldwide [1]. One of the most commonly affected genes in cancer is the proto-oncogene Ras. In fact, mutations that elevate its activity are present in ∼30% of human cancers and give rise to some of the most aggressive tumors [2]. However, hyperactivation of Ras signaling alone is insufficient to produce malignancy (reviewed in [3]. Additional mutations in other genes are required for Ras-driven malignant tumorigenesis. Thus, identifying genes that modulate the oncogenic capacity of Ras is imperative in our fight against cancer. Of particular interest are factors important for oncogenesis but that are dispensable for normal development and homeostasis. These molecules are ideal targets for cancer therapy, as they hold the potential to prevent the growth of cancer cells while having little or no effect on normal tissues.

True Ras proteins (H-, N- and K-RAS in humans and DRas1 in *Drosophila*) connect activated tyrosine kinase receptors, such as EGFR and FGFR, to intracellular transducers. Upon ligand binding, EGFR dimerizes and is activated via trans-phosphorylation, which leads to the recruitment of signaling molecules with Src homology 2 (SH2) domains, such as Grb2 and Shc (reviewed in [4]. These proteins mediate the recruitment of the guanine-nucleotide exchange factor (GEF) Son of Sevenless (Sos) to the receptor complex, which results in Ras activation [5,6]. Similar to Ras, EGFR is also mutationally-activated and/or overexpressed in one-fifth of all human cancers [7]. Likewise, even though EGFR expression is altered in many epithelial tumors, cooperation with other oncogenic lesions is required for malignant transformation and invasion [8,9].

The EGFR/Ras signaling pathway has been systematically studied during *Drosophila* embryonic and imaginal disc development, where it participates in the regulation of cell proliferation, growth, differentiation, migration and survival (reviewed in [10]. A particular contribution of the fly model has been the identification of positive and negative regulators through genetic screens. Some of these components are transcriptionally-regulated by the pathway, generating both positive and negative feedback loops (reviewed in [10]). For example, the expression of *rhomboid*, a gene encoding a serine protease that cleaves the ligand Spitz, is activated by EGFR signaling to further increase pathway activity. On the other hand, proteins encoded by EGFR-activated genes *kekkon*, *argos*, *sprouty* and *MKP3* antagonize the pathway at different levels (reviewed in [11,12]).

The *Drosophila* wing is a very convenient system to analyze EGFR signaling, because of its simplicity and our ability to alter and visualize pathway activity *in situ* with relative ease. The wing develops from a relatively flat epithelial sac, known as the wing imaginal disc, which grows by cell proliferation during larval stages and differentiates during pupal development

into the fly thorax and wing. The EGFR pathway is essential for development of the wing imaginal disc, acting to promote the early proximo-distal patterning of the disc and the formation of wing veins and sensory organs at later stages [13–16]. In this and other developmental contexts, EGFR signaling is tightly regulated by a carefully orchestrated spatial and temporal distribution of activating and inhibitory factors (reviewed in [17]).

The wing disc also offers an effective system to study tumor progression and oncogenic cooperation (reviewed in [18]). The induction of cell mosaics over-expressing activated Ras (Ras$^{V12}$) or EGFR (λTop) in the wing disc gives rise to hyperplastic growth [19]. This has been exploited in genetic screens to identify additional genes that can either suppress or enhance the growth of cells overexpressing Ras$^{V12}$ or λTop mutations. EGFR/Ras pathway interactors include genes regulating apico-basal cell polarity, RhoGEF, genes encoding lysosomal proteins, proteins involved in mitochondrial respiratory function and several microRNAs (reviewed in [20]). Intriguingly, one of the genes found to cooperate with oncogenic Ras, both in *Drosophila* and in human cells, is EGFR itself [21–24]. However, in this context, it is noteworthy that none of the factors regulating EGFR during normal development have yet been identified as oncogenic EGFR/Ras cooperating genes.

In this work, we have identified the gene *CG33993*, which we have named *EGFRAP* for <u>EGFR</u> <u>a</u>daptor <u>p</u>rotein, as a novel modulator of oncogenic Ras in the *Drosophila* wing imaginal disc. Elimination of *EGFRAP*, either by mutation or RNAi, enhances *Ras$^{V12}$*-mediated tissue hyperplasia. *EGFRAP* encodes a protein containing a conserved SH2 domain, which physically interacts with the active form of EGFR. EGFRAP localizes to the apical region of cells with high levels of EGFR activity, such as wing margin cells. The apical accumulation of EGFRAP in wing margin cells is regulated by the Notch pathway. In addition, we found that ectopic activation of the EGFR/Ras pathway in wing discs drives EGFRAP expression in the pouch territory. This EGFR/Ras-driven ectopic EGFRAP expression depends on Notch signaling, which is itself upregulated in response to EGFR hyperactivation. Based on these results, we propose that *EGFRAP* is part of a novel negative feedback loop acting as an important regulator of cell proliferation driven by excessive EGFR/Ras signaling. EGFRAP mutations are viable and don´t display EGFR-related phenotypes, suggesting that EGFRAP function is not required during normal development. However, we find that EGFRAP and its homologous gene PVRAP act redundantly in normal cells to restrain EGFR/Ras signaling. EGFRAP specificity for cells with high EGFR activity is consistent with activation of a Notch pathway-mediated negative feedback loop, where it acts to help prevent excessive EGFR activity and uncontrolled cell growth.

## Results

### *EGFRAP* knockdown enhances Ras$^{V12}$ hyperplastic phenotype

Ectopic expression of activated *Ras* (*Ras$^{V12}$*) in *Drosophila* wing imaginal discs produces hyperplasia due to increased cell growth, accelerated G1-S transition and cell shape changes [25,26]. To isolate new modulators of *Ras$^{V12}$* activity, we used RNAi to knockdown a battery of candidate genes and search for knockdowns that enhanced the *Ras$^{V12}$* phenotype. Ectopic expression of *Ras$^{V12}$* in the dorsal compartment of wing imaginal discs, by means of *apterous*-Gal4 (*ap>*GFP; *Ras$^{V12}$*), or in a discrete stripe along the A/P boundary, using a *tub-Gal80$^{ts}$; patched*-Gal4 combination (*ptc$^{80ts}$>* GFP; *Ras$^{V12}$*), induces overgrowth of the tissue and the formation of ectopic folds [26] (Fig 1A–1B', S1A–S1B' and S1D Fig). Using this model to screen for enhanced Ras hyperplasia, we identified the gene *EGFRAP*. Although *EGFRAP* RNAi had no detectable effect in wild-type cells (*ap>*GFP; *EGFRAP$^{RNAi}$*, Fig 1C and 1C'), it enhanced the phenotype of *Ras$^{V12}$* cells (*ap>*GFP; *Ras$^{V12}$*; *EGFRAP$^{RNAi}$*, Fig 1D and 1D' and *ptc$^{80ts}$>*GFP; *Ras$^{V12}$*; *EGFRAP$^{RNAi}$*, S1C, S1C' and S1D Fig).

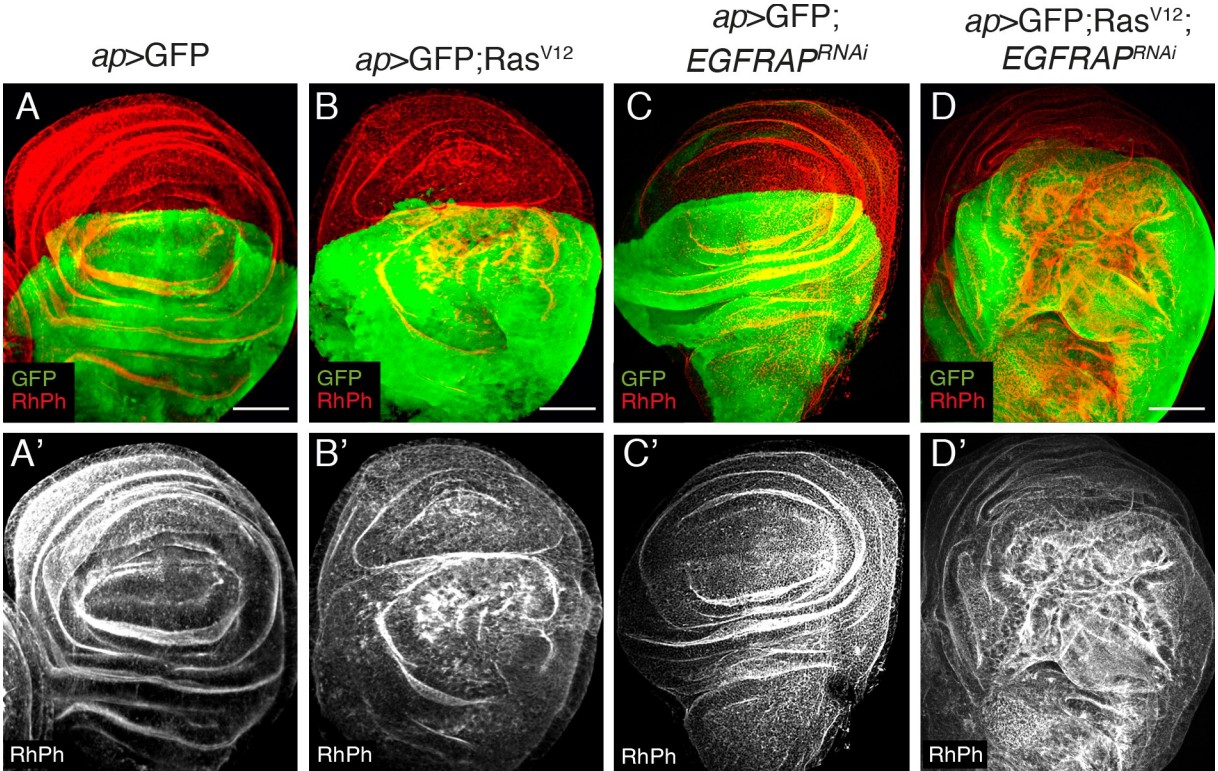

**Fig 1. *EGFRAP* Knock down enhances Ras<sup>V12</sup>-dependent tissue hyperplasia in *Drosophila* wing imaginal discs.** (A-D) Maximal projection of confocal views of wing imaginal discs from third-instar larvae expressing GFP (green) and the indicated UAS transgenes under the control of *apterous* Gal4 (*ap-Gal4*), stained with anti-GFP (green) and Rhodamine Phalloidin to detect F-actin (RhPh, red) (A'-D'). Scale bars, 50 μm (A-D).

The formation of additional folds could be due to increased cell proliferation, changes in cell shape or both. Staining with an antibody against phosphorylated Histone H3 (PH3) revealed that, in agreement with previous results [26,27], Ras$^{V12}$ overexpressing discs showed a reduction in PH3$^+$ cells (S2A–S2B' and S2D Fig). This was not affected by *EGFRAP$^{RNAi}$* co-expression (S2C and S2D Fig). To analyze possible cell shape changes, cell plasma membranes were labelled with a membrane-targeted fluorescent protein (myristoylated Tomato, myrT, Fig 2A–2I). The wing pouch cells in late third-instar control discs are elongated and columnar, with a mean height of 12.6 μm (Fig 2A' and 2J), an apical area of 4.5 μm$^2$ (Fig 2D and 2K) and a basal area of 7.1 μm$^2$ (Fig 2G and 2L). In contrast, *Ras$^{V12}$* expressing cells are shorter and more cuboidal, with a mean height of 7.8 μm (Fig 2B' and 2J), an apical area of 14.1 μm$^2$ (Fig 2E and 2K) anda basal area of 26.2 μm$^2$ (Fig 2H and 2L). If we consider disc cells as truncated prisms, the resulting volume of control and *Ras$^{V12}$* cells would be around 72.46 μm$^3$ and 154.75 μm$^3$, respectively. This significant difference in cell volume is in agreement with previous results showing that *Ras$^{V12}$* cells show increased cellular growth [25,26]. *EGFRAP* knockdown enhanced the expansion of apical (21.6 μm$^2$; Fig 2F and 2K) and basal (30.7 μm$^2$; Fig 2I and 2L) areas, with cell volume increasing from 154.75 μm$^3$ to 187.32 μm$^3$. Given the constraints imposed by the peripodial membrane, the observed cell shape changes and the increase in cell size could explain the formation of extra folds.

Occasionally, some GFP positive cells were found outside of the dorsal domain in both *ap>GFP; Ras$^{V12}$* and *ap>GFP; Ras$^{V12}$; EGFRAP$^{RNAi}$* expressing discs, with the latter having the stronger phenotype (S3A–S3C' and S3E Fig). Invading cells were always found basally and

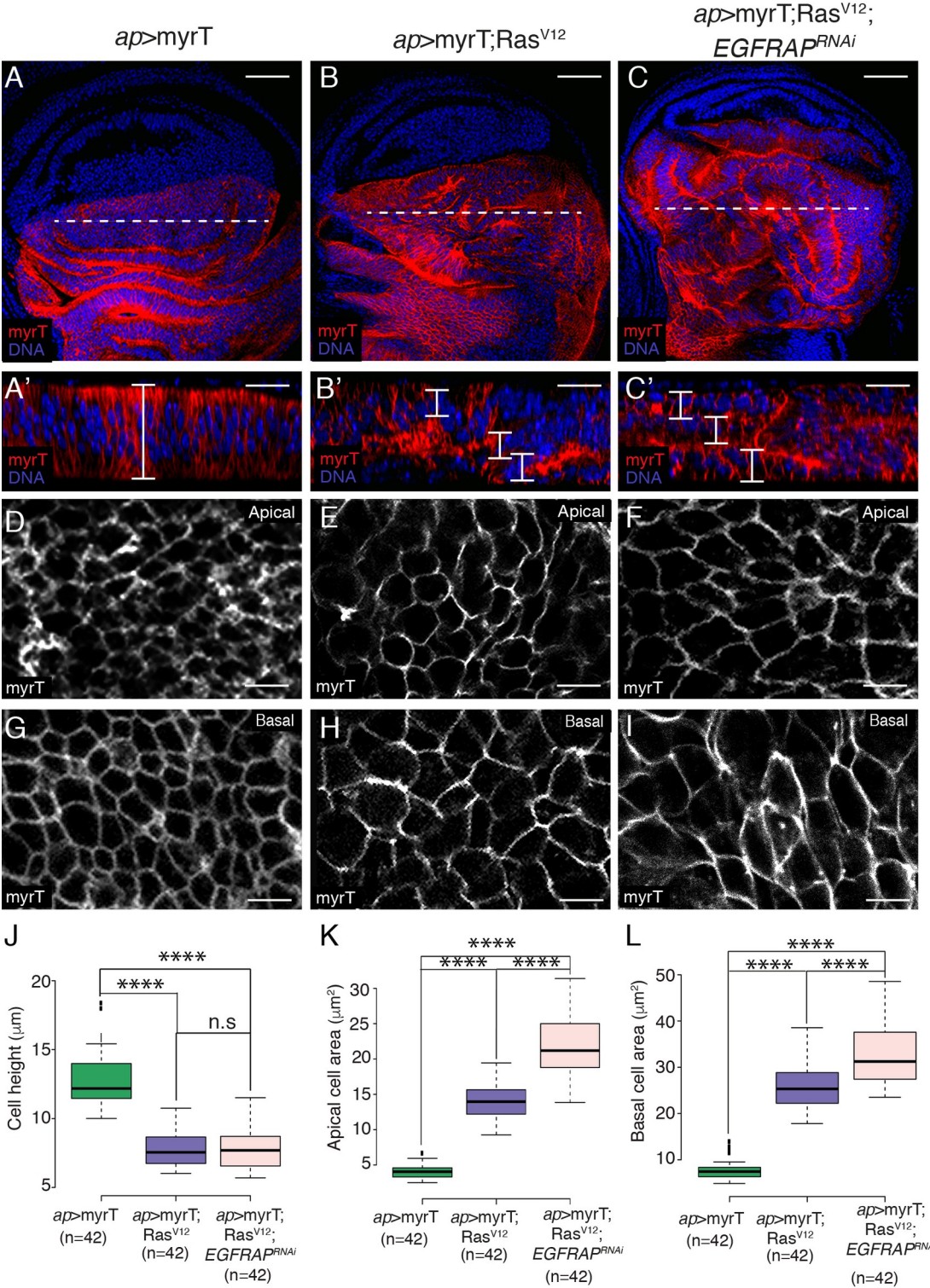

**Fig 2. *EGFRAP* knockdown increases Ras^V12^-dependent cell shape changes and growth in *Drosophila* wing imaginal discs.**

**Fig 2. *EGFRAP* knockdown increases Ras$^{V12}$-dependent cell shape changes and growth in *Drosophila* wing imaginal discs.**
(A-C) Maximal projection of confocal images of wing imaginal discs from third-instar larvae expressing myristoylated-Tomato (MyrT, red) and the indicated UAS transgenes under the control of *ap*-Gal4, stained with anti-Tomato (red) and the nuclear marker Hoechst (DNA, blue). (A'-C') Confocal *xz* sections along the white dotted lines of wing discs shown in A-C. The apical side of wing discs is at the top. Brackets indicate cell height. (D-F) Apical and (G-I) basal surface views of the indicated genotypes. (J-L) Box plots of the cell height (J), apical cell area (K) and basal cell area (L) of the indicated genotypes. The statistical significance of differences was assessed with a t-test, ****P value<0.0001. Scale bars, 50 μm (A-C), 30 μm (A'-C') and 10 μm (D-I).

most of them were positive for the apoptotic marker, cleaved Dcp-1 (white arrow in S3D–S3D" Fig). Effector caspases are active in tumors and are associated with metastasis [28]. Furthermore, caspase activity has been proposed to direct the migration of transformed cells in wing imaginal discs [29]. In this context, we found that co-expression of Diap1, an inhibitor of apoptosis, blocked the invasiveness of both $Ras^{V12}$ and $Ras^{V12}$; $EGFRAP^{RNAi}$ cells (S3E Fig). These results suggest that $EGFRAP$ downregulation might cooperate with $Ras^{V12}$ to promote tumor invasion associated with effector caspase activity.

Previous studies have shown that overexpression of activated Ras stimulates JNK and promotes the death of nearby wild-type cells [30,31]. In agreement with this, we detected a clear enrichment of apoptosis (Fig 3B, 3B' and 3D, n = 18, and white arrowhead in S3D Fig) and JNK activity (Fig 3F and 3H, n = 19) in wild-type (GFP negative) ventral cells located at the D/V boundary in $ap$>GFP; $Ras^{V12}$ discs compared to controls (Fig 3A, 3A', 3D, 3E and 3H, n = 22). Co-expression of $EGFRAP^{RNAi}$ resulted in an increased number of apoptotic cells (Fig 3C, 3C', 3D, 3G and 3H, n = 20). Cell death can induce compensatory proliferation [32]. This could explain the accumulation of PH3$^+$ cells observed at the D/V boundary of $ap$>GFP; $Ras^{V12}$ and $ap$>GFP; $Ras^{V12}$; $EGFRAP^{RNAi}$ discs (S2B–S2C' and S2D Fig). Apoptosis of the wild-type neighbors of $Ras^{V12}$ cells has also been explained by an increase in tissue compaction due to the enhanced growth of mutant cells [33]. Here, we find that the wild-type ventral region of $ap$>GFP; $Ras^{V12}$ discs (Fig 3I) were more compressed than in $ap$>GFP (Fig 3I) discs. This phenotype was further enhanced in $ap$>GFP; $Ras^{V12}$; $EGFRAP^{RNAi}$ wing discs (Fig 3K).

Although ectopic $Ras^{V12}$ expression in wing disc cells alone does not affect cell polarity ([34]; S4A–S4B" and S4D–S4E" Fig), the removal of polarity genes enhances the hyperplastic phenotype of $Ras^{V12}$ [35]. Thus, we tested whether cell polarity was affected in $Ras^{V12}$; $EGFRAP^{RNAi}$ disc cells. We found that downregulation of $EGFRAP$ did not alter the polarity of $Ras^{V12}$ cells (S4C–S4C" and S4F–S4F" Fig).

All together, these results suggest that $EGFRAP$ modulates $Ras^{V12}$-mediated tissue hyperplasia by enhancing cell shape changes and cellular growth.

## CRISPR/Cas9 mediated generation of *EGFRAP* mutant alleles

To further characterize the role of $EGFRAP$ as a modulator of $Ras^{V12}$-mediated hyperplasia, we used CRISPR/Cas9 to generate specific $EGFRAP$ alleles (see Materials and Methods). The $EGFRAP$ gene encodes two isoforms, a full-length long isoform ($EGFRAP$-RA) and a short isoform ($EGFRAP$-RB), whose transcription start site maps to the beginning of exon 3 (Fig 4A). We generated two $EGFRAP$ mutant alleles, in which both isoforms ($EGFRAP^{L/S}$) or only the long isoform ($EGFRAP^L$) were truncated (Fig 4A). The $EGFRAP^{L/S}$ allele truncates 90.7% of the short isoform and 67% of the long isoform (Fig 4A). The $EGFRAP^L$ allele truncates 78.2% of the long isoform (Fig 4A). In addition, as exon 5 of $EGFRAP$ encodes a conserved SH2 domain known to interact with phosphorylated tyrosines [36], we also generated a mutant allele whose SH2 domain was completely eliminated ($EGFRAP^{\Delta SH2}$; Fig 4A).

All $EGFRAP$ mutant alleles were homozygous viable and displayed no obvious morphological abnormalities, indicating that $EGFRAP$ is dispensable for viability in *Drosophila*. To confirm the role of $EGFRAP$ as a modulator of $Ras^{V12}$-mediated hyperplasia, we tested for synergetic interactions between $EGFRAP$ mutations and $Ras^{V12}$ in wing imaginal discs (Fig 4 and S5 Fig). We found that expression of $Ras^{V12}$ in the posterior compartment of $EGFRAP$ mutant discs (Fig 4D–4D" and S5B–S5C" Fig) resulted in an enhanced folding phenotype, similar to that observed in $Ras^{V12}$; $EGFRAP^{RNAi}$ discs (Fig 1D and 1D'). The strong $Ras^{V12}$ enhancement by $EGFRAP^L$ and $EGFRAP^{\Delta SH2}$ alleles suggests that the function of $EGFRAP$ as a

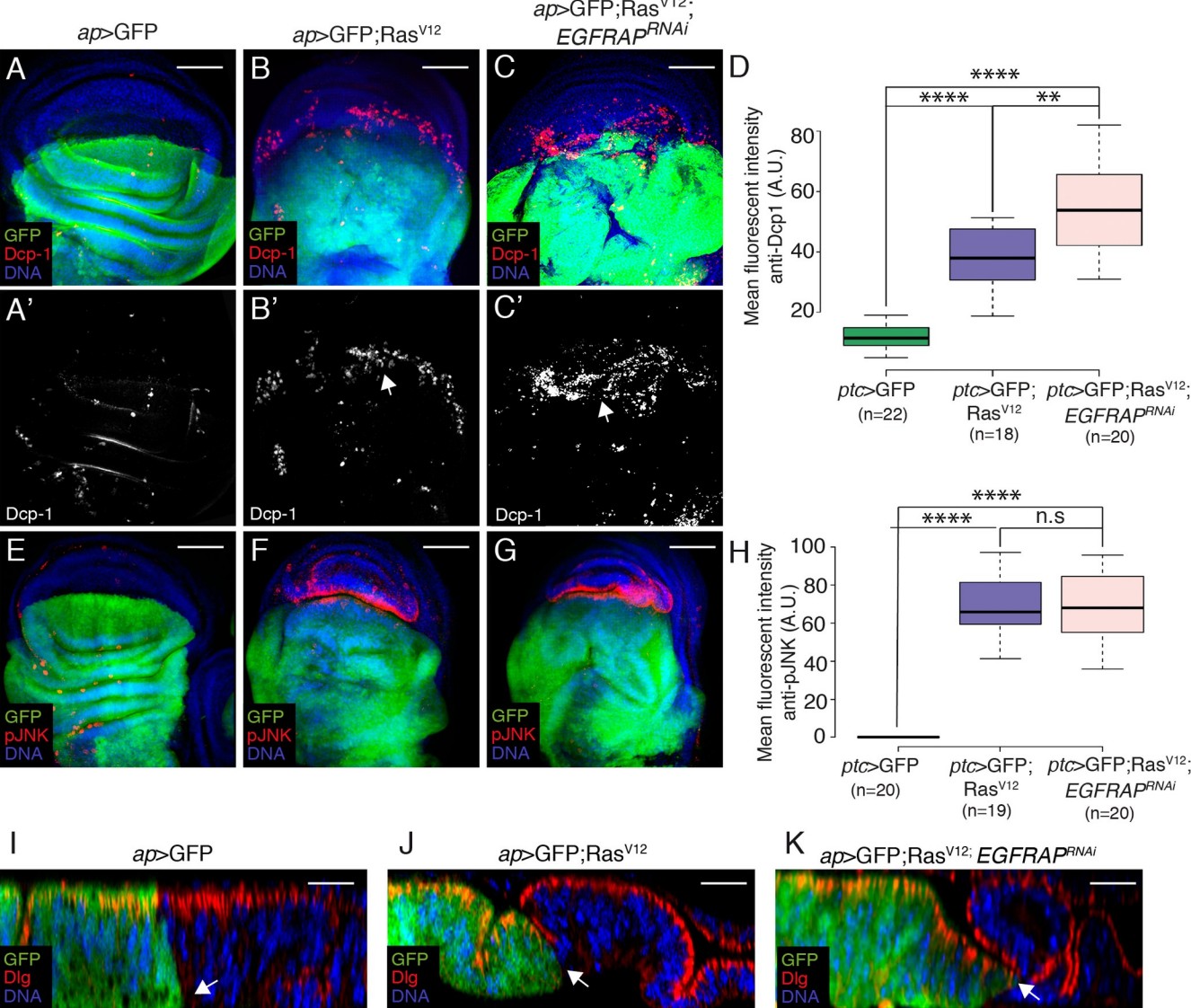

**Fig 3. *EGFRAP* downregulation increases apoptosis and JNK activity in wild-type cells adjacent to Ras^V12 expressing cells.** (A-G) Maximal projection of confocal views of wing imaginal discs from third-instar larvae expressing GFP (green) and the indicated UAS transgenes under the control of *ap*-Gal4, stained with anti-GFP (green), anti-Dcp-1 (red in A-C, white in A'-C'), anti-pJNK (red in E-G) and Hoechst (DNA, blue). (D, H) Box plots of mean fluorescent Dcp-1 (D) and pJNK (H) intensities of wing discs of the designated genotypes. (I-K) Confocal *yz* sections parallel to the A/P axis of wing imaginal discs from third-instar larvae expressing GFP (green) and the indicated UAS transgenes under the control of *ap*-Gal4, stained with anti-GFP (green), anti-Dlg (red) and Hoechst (DNA, blue). Apical side of wing discs is to the top. Arrows in I-K point to the border between dorsal experimental and ventral control cells. The statistical significance of differences was assessed with a t-test, **** and *** P values <0.0001 and <0.001, respectively. Scale bars, 50 μm (A-G) and 30 μm (I-K).

modulator of *Ras^V12*-mediated hyperplasia is performed mainly by the long isoform and that the SH2 domain is crucial for this function.

The absence of a loss-of-function phenotype for *EGFRAP* could be due to compensation by other adaptor proteins performing similar functions. We noticed that adjacent to *EGFRAP* lies the gene *PVRAP*, which also encodes an SH2 domain. Furthermore, the amino acid sequences of their SH2 domains share 74% identity, versus 26%-35% identity for pairwise comparisons of *EGFRAP* with SH2 domains found in other *Drosophila* proteins. We found that RNAi knockdown of *PVRAP* in wing imaginal discs enhanced the Ras^V12 overgrowth and folding

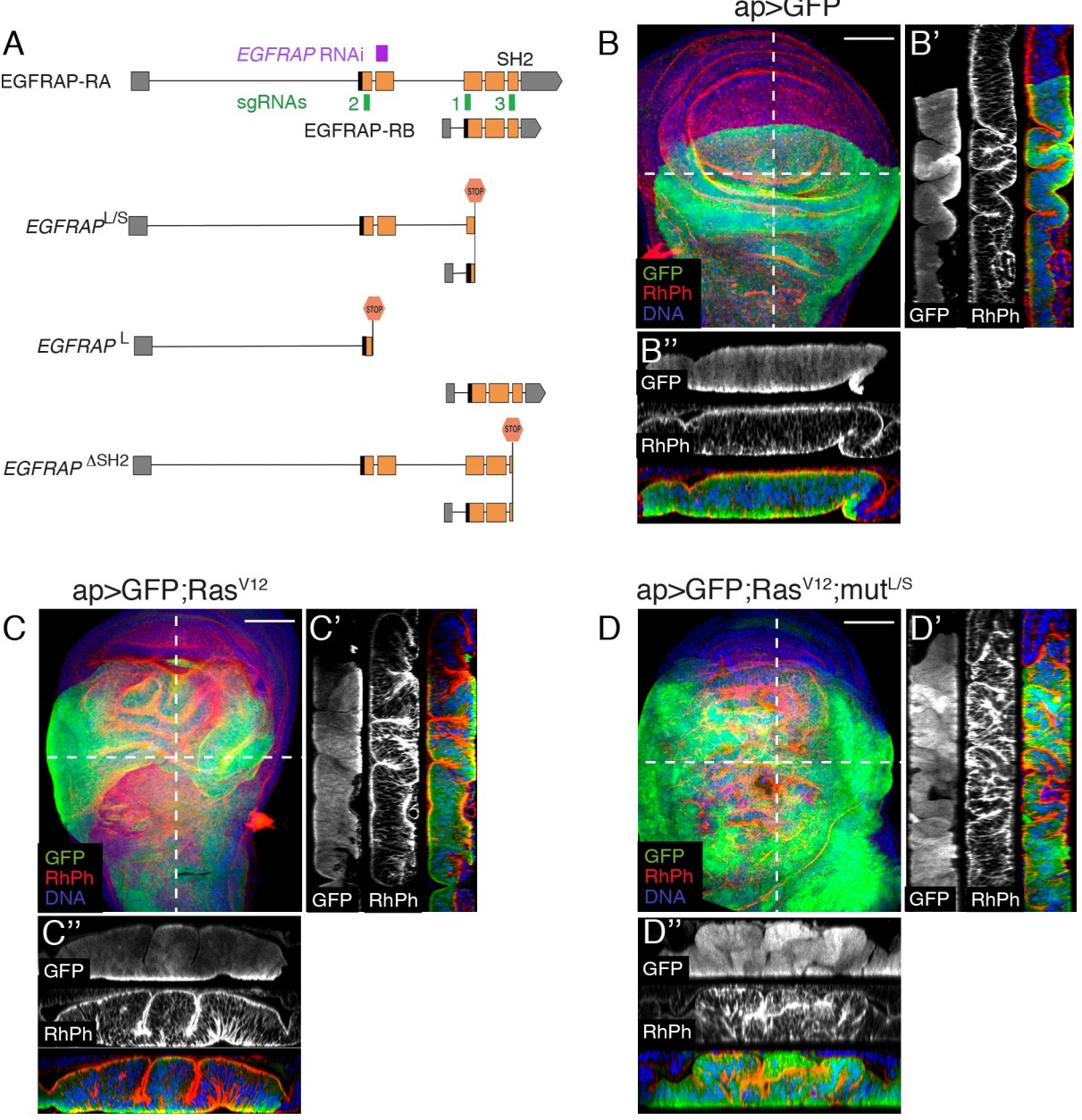

**Fig 4. Generation of *EGFRAP* mutant alleles supports its role as modulator of Ras$^{V12}$-mediated tissue hyperplasia.** (A) Schematic representation of the *EGFRAP* locus (3$^{rd}$ chromosome), *EGFRAP* mutants generated, sgRNAs used for generation of mutants (green boxes 1–3) and sequence targeted by *EGFRAP$^{RNAi}$* construct (purple box). (B-D) Maximal projection of confocal images of wing imaginal discs from third-instar larvae of the indicated genotypes stained with anti-GFP (green), RhPh (red) and Hoechst (DNA, blue). (B'-B", C'-C" and D'-D") Confocal sections of wing discs of the specified genotypes along the white dotted lines shown in B, C and D, respectively, parallel (B', C' and D') or perpendicular (B", C" and D") to the A/P border. Apical sides of wing discs are to the left (B', C' and D') or to the top (B", C" and D"). Scale bars, 60 μm (B-D).

phenotype (Fig 5A–5C'), in a similar way to *EGFRAP*. Furthermore, downregulation of *PVRAP* enhanced the effects of eliminating *EGFRAP* in Ras$^{V12}$ expressing wing discs (ap>GFP; Ras$^{V12}$; PVRAP$^{RNAi}$; EGFRAP$^{L/S}$; Fig 5D and 5D'). In addition, while downregulation of either *EGFRAP* (ap>EGFRAP$^{RNAi}$) or *PVRAP* (ap>PVRAP$^{RNAi}$) on their own did not cause any visible phenotype in the adult (Fig 5E–5G and 5I–5K), the simultaneous

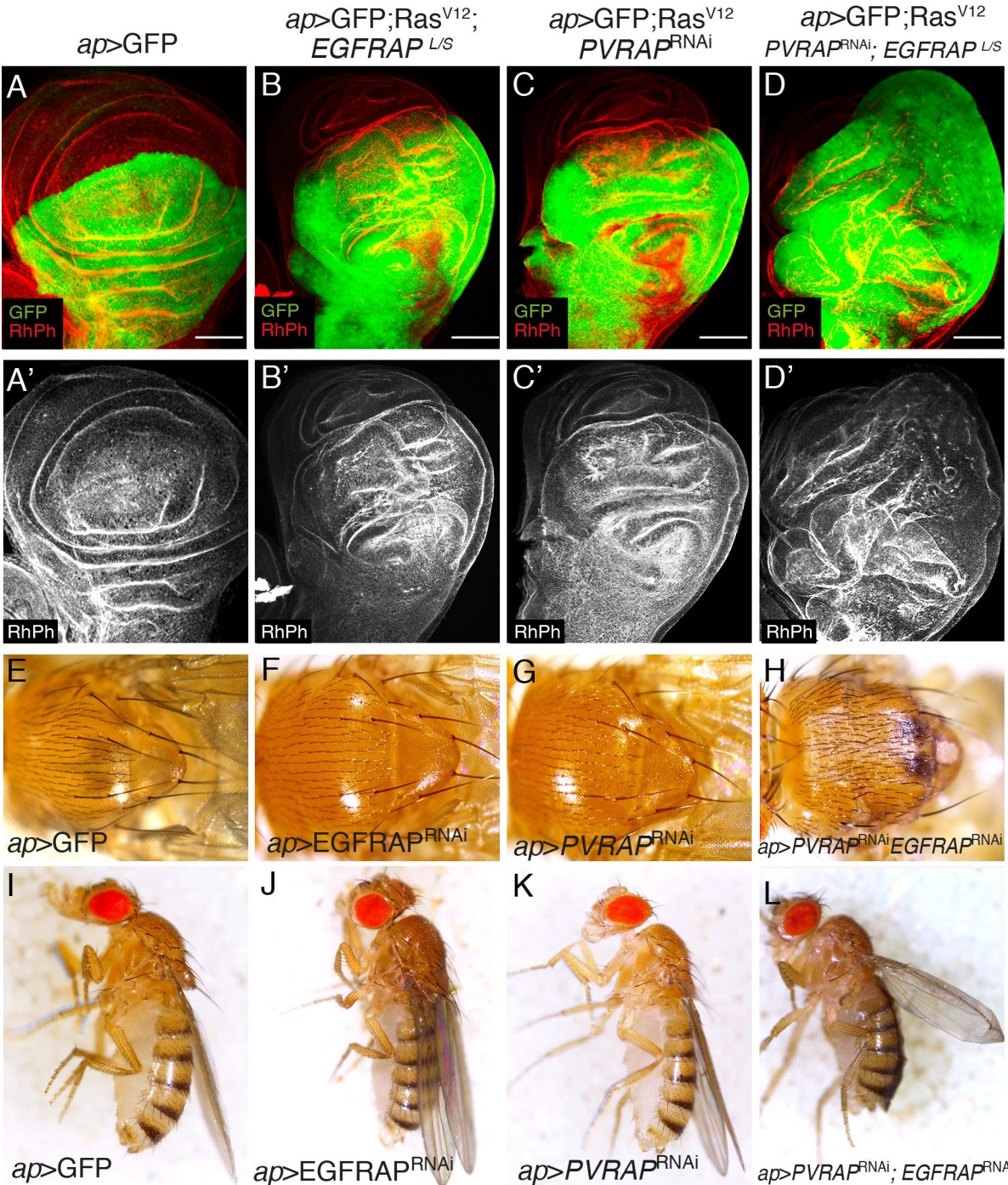

**Fig 5. _PVRAP_ downregulation enhances _EGFRAP_ loss of function phenotypes.** (A-D') Maximal projection of confocal images of third instar wing imaginal discs expressing the indicated UAS transgenes under the control of _ap_-Gal4 stained with anti-GFP (green) and RhPh (red). (E-H) Dorsal view of _Drosophila_ noti of the specified genotypes. (I-L) Images of female flies of the indicated genotypes. Scale bars, 50 μm (A-C).

downregulation of both genes (_ap>EGFRAP^RNAi; PVRAP^RNAi_) resulted in ectopic sensory organs in the notum (Fig 5H), a phenotype that resembles increased EGFR activity [14], and blisters in the wing (Fig 5L).

## EGFRAP is expressed in wing margin cells and co-localizes apically with EGFR

To analyze the expression pattern of *EGFRAP* in wing discs, we carried out *in situ* hybridization with a labelled probe against *EGFRAP* mRNA (see Materials and Methods). *EGFRAP* transcripts were observed at high levels along two parallel stripes of cells on either side of the wing pouch D/V boundary (Fig 6A). We also noticed a region straddling the A/P boundary with slightly lower expression levels than the rest of the wing margin (Fig 6A). This distribution along the D/V boundary was similar to that described for reporters of EGFR/Ras pathway activation [37]. *EGFRAP* expression was also observed in other discrete regions of the wing imaginal disc outside of the pouch (Fig 6A). To extend our analysis of *EGFRAP* expression, we generated an antibody that recognizes both long and short EGFRAP isoforms (see Materials and Methods). Immunostaining of wild-type wing discs with this antibody showed an expression pattern similar to that of *EGFRAP* mRNA along the wing margin, with high levels in two stripes of cells on either side of the D/V boundary (Fig 6B and 6C) and lower levels in a small region at the A/P boundary (Fig 6B and 6C). The expression of EGFRAP in cells around the wing margin disappeared in EGFRAP$^{L/S}$ mutant wing discs (S5D Fig) and in the posterior region of wing discs expressing *EGFRAP$^{RNAi}$* under the control of *engrailed*-Gal4 (en>*EGFRAP$^{RNAi}$*, S5E Fig), demonstrating the specificity of our antibody. EGFR has been found to localize apically in imaginal disc cells [38]. If EGFRAP were to interact with EGFR in these cells, we would expect the two proteins to co-localize. To test our hypothesis, we immunostained wing discs from transgenic flies carrying a superfolder-GFP inserted into the EGFR locus (EGFR-sfGFP, [39] with anti-GFP and anti-EGFRAP antibodies. We found that EGFR-sfGFP co-localized with EGFRAP in the apical region of cells with high levels of EGFRAP (Fig 6D and 6D'), suggesting that indeed EGFRAP could interact with EGFR. This was in agreement with results from a previous yeast two-hybrid screen, using the intracellular domain of EGFR as bait, where we identified a partial clone coding for the EGFRAP protein [40] (see Materials and Methods and S6A Fig). The isolated clone encoded a complete SH2 domain plus the C-terminal end of EGFRAP. Here, by using a battery of baits previously described [40] (see Materials and Methods and S6A Fig), we found that EGFRAP bound strongly to wild-type EGFR (S6B Fig). We also found that this interaction was dependent on the presence of tyrosine residues in the EGFR cytoplasmic tail, suggesting that interaction with EGFRAP is modulated by tyrosine phosphorylation (S6 Fig).

The expression of many modulators of the EGFR/Ras pathway in the wing pouch is itself regulated by the EGFR/Ras pathway. To test whether this was also the case for EGFRAP, we analyzed its expression in wing discs expressing either activated (λtop, [41]) or dominant negative (EGFR$^{DN}$, [42]) forms of EGFR in the dorsal compartment. We found that ectopic expression of λtop induced ectopic EGFRAP expression in the wing pouch (Fig 6E–6F"). However, we observed that downregulation of EGFR did not affect EGFRAP expression (Fig 6H). Note that in this case we used *hedgehog*-Gal4, which drives expression in the posterior compartment [43]. The specific expression of EGFRAP in cells flanking the D/V boundary and unresponsiveness to decreased EGFR activity suggest that other factors also regulate EGFRAP expression in the wing pouch.

The EGFRAP expression pattern resembles that of Senseless, a transcription factor regulated by wingless signaling, downstream of the Notch pathway [44,45]. Thus, we tested if EGFRAP expression was regulated by Notch. We found that RNAi knockdown targeting Notch (*hh*>GFP; N$^{RNAi}$, Fig 6I and 6J) reduced EGFRAP levels. Conversely, ectopic expression of an active intracellular fragment of the Notch receptor (*ap*>GFP; N$^i$, Fig 6K) increased EGFRAP expression in the wing pouch (Fig 6K). Furthermore, we found that downregulation

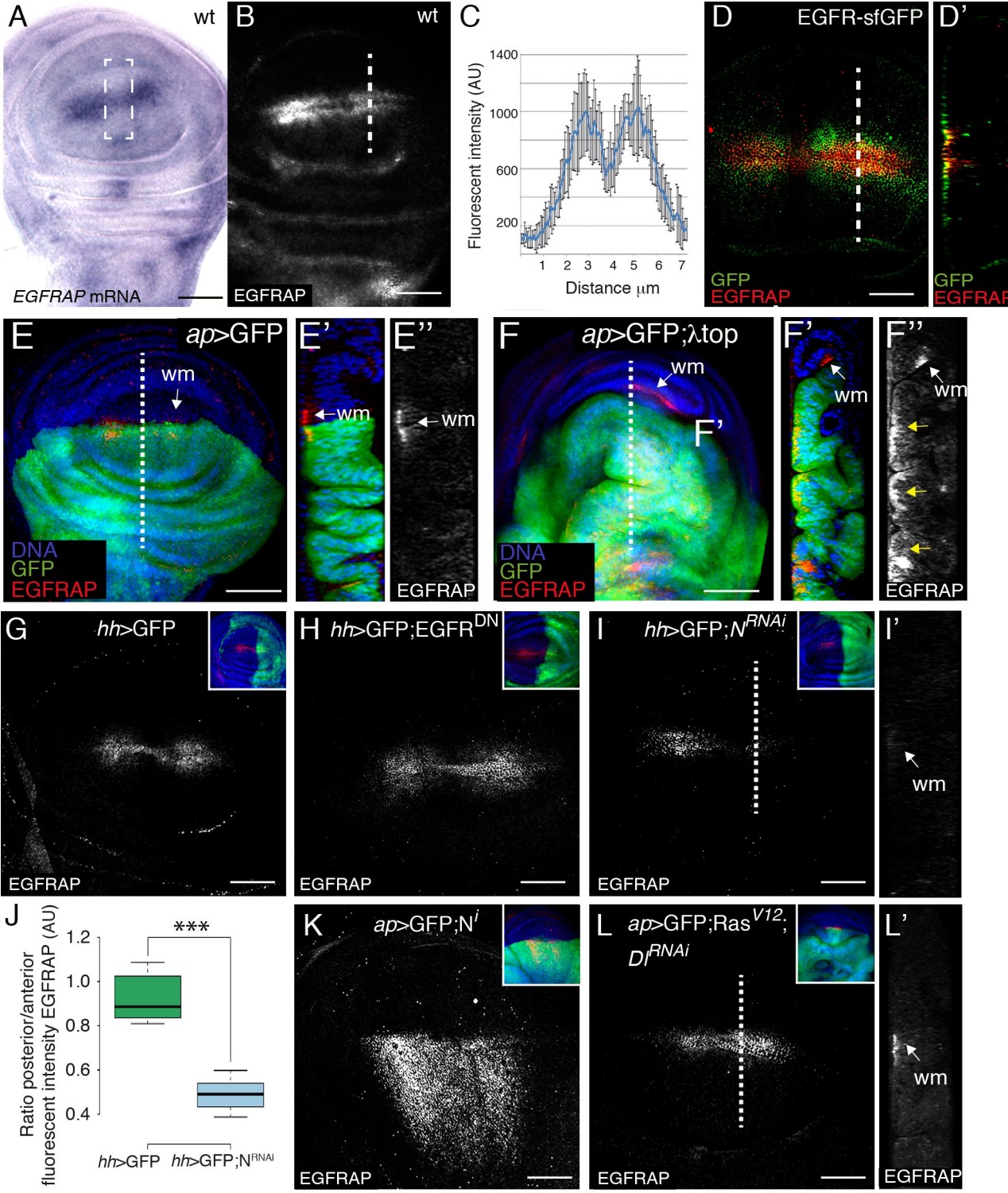

**Fig 6. Expression of EGFRAP in *Drosophila* wing discs.** (A) *In situ* hybridization of a third instar wing imaginal disc with a probe for *EGFRAP* mRNA. (B) Maximal projection of confocal images of a wild-type (wt) third instar wing imaginal disc stained with anti-EGFRAP. (C) Quantification of EGFRAP levels in the region of the white dotted line in (B). (D) Maximal projection of confocal images of an EGFR-sfGFP third instar wing imaginal disc stained with anti-EGFRAP (red) and anti-GFP (green). (D') Confocal *yz* section, along the white dotted line in D. (E, F) Maximal projection of confocal views of wing discs expressing the indicated UAS transgenes under the control of *ap*-Gal4 stained with anti-GFP (green), anti-EGFRAP (red) and Hoechst (DNA, blue). (E'-E" and F'-F") Confocal *yz* sections along the white dotted lines shown in E and F, respectively. Apical side of wing discs is to the left. White arrows indicate wing margin cells (wm). (G, H, I, K, L) Maximal projection of confocal images of wing discs expressing GFP and

the specified UAS transgenes under the control of *hedgehog*-Gal4 (*hh-Gal4*) (G-I) and *ap*-Gal4 (K, L), stained with anti-EGFRAP (red), anti-GFP (green) and Hoechst (DNA, blue). (J) Box plot of the posterior/anterior EGFRAP intensity ratio of wing discs of the designated genotypes. (I', L') Confocal *yz* sections along the white dotted lines are shown in I and L, respectively. Apical side of wing discs is to the left. The statistical significance of differences was assessed with a t-test, *** P value<0.001. wm (wing margin). Scale bars, 60 μm (A, B) and 50 μm (D, E, F, G, H, I, K, L).

of *Delta* was sufficient to rescue the increase in EGFRAP levels due to EGFR pathway overactivation (Fig 6L).

EGFR is also expressed and required for the proliferation, survival and differentiation of most cell types in the *Drosophila* eye disc, including all photoreceptors (R1-R7), cone and pigment cells. Accordingly, we tested whether EGFRAP was also expressed in the eye disc. Interestingly, we found that EGFRAP was only expressed in a subset of the cells requiring EGFR activity (S7A–S7B' Fig). More specifically, using the Hedgehog-LacZ marker (Hh-LZ,), which is expressed in photoreceptors R2-R5 (S7C and S7C' Fig, [46]), and an anti-Senseless (Sens) antibody, which labels R8 (S7D and S7D' Fig, [47]), we could determine that the photoreceptor expressing high EGFRAP levels was R7. In addition, ectopic expression of λtop in eye disc clones (hs>GFP; λtop) induced ectopic expression of EGFRAP (S7E and S7E' Fig). Thus, EGFRAP expression and its relationship with EGFR during *Drosophila* eye development appears analogous to the situation in wing imaginal discs.

### *EGFRAP* regulates EGFR activity

To further test whether *EGFRAP* acts as a negative regulator of the EGFR/Ras pathway, we altered *EGFRAP* expression levels and monitored the effect on pathway activity by visualizing pERK accumulation. We found that reducing (*en>EGFRAP$^{RNAi}$*) EGFRAP expression levels in the posterior compartment of wing discs, increased pERK levels (Fig 7A–7B' and 7D). To increase *EGFRAP* levels, we used a transgenic line expressing a GFP-tagged version of the protein (*UAS*-EGFRAP-GFP; see Materials and Methods). In this case, we found that by increasing EGFRAP expression levels (*en>RFP;* EGFRAP-*GFP*), pERK expression was reduced (Fig 7C and 7D). We also found that *EGFRAP* overexpression in the dorsal region of the wing disc resulted in the loss of notum and wing tissue, a phenotype similar to that observed upon reduction of EGFR/Ras activity (Fig 7E and 7F) [48]. Other SH2 domain-containing negative regulators of EGFR act by promoting the internalization and degradation of the receptor. However, we found that *EGFRAP* overexpression did not affect the levels or localization of EGFR-sfGFP (Fig 7G). These results suggest that the phenotypes associated with reduced EGFR signaling mediated by *EGFRAP* overexpression are not a consequence of defects in receptor trafficking.

We also found that flies overexpressing *EGFRAP* in the developing eye discs exhibited smaller rough eyes (S7F and S7G Fig), a phenotype that mimics overexpression of other negative regulators of EGFR, such as *argos* [49].

Altogether, our data suggest that *EGFRAP* acts as a negative feedback regulator of excessive EGFR/Ras signaling, as EGFR/Ras pathway activation results in an increase in EGFRAP expression, which in turn limits EGFR activity. Supporting this hypothesis, we found that downregulation of EGFRAP enhanced the folding phenotype due to expression of λtop (S8 Fig).

## Discussion

EGFR signaling plays important roles during multicellular development and tissue homeostasis, affecting cell division and growth, cell fate choices, cell viability and organ morphogenesis. During morphogenesis or in normal differentiated tissues, cells are not persistently exposed to high concentrations of activating epidermal growth factors. However, over recent years it has been found that this scenario can change in conditions of pathway hyperactivation, where cells have to impose

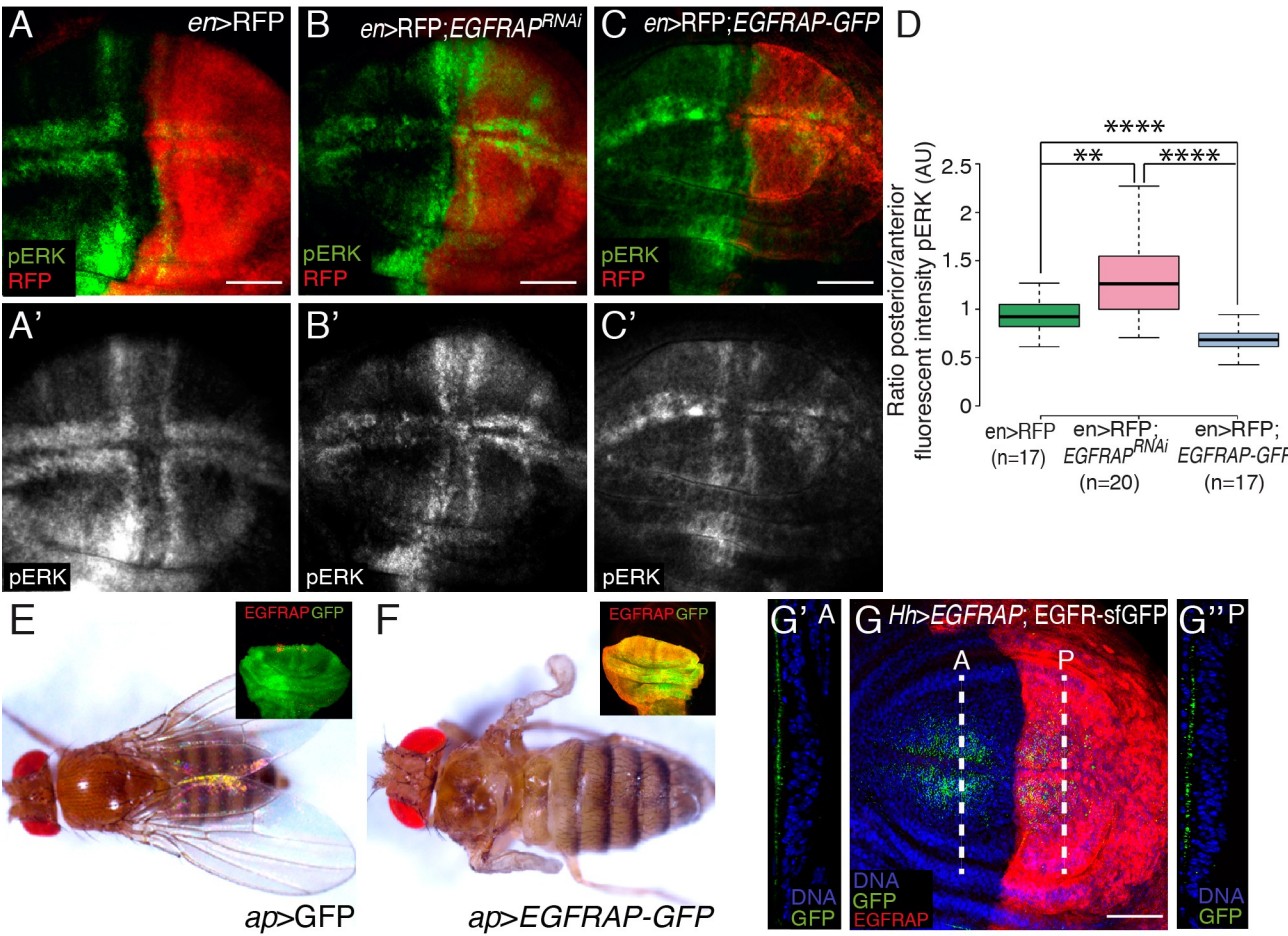

**Fig 7. *EGFRAP* regulates EGFR activity in *Drosophila* wing imaginal discs.** (A-C') Maximal projection of confocal images of third instar wing imaginal discs expressing the indicated UAS transgenes under the control of *en*-Gal4, stained with anti-RFP (red) and anti-pERK (green). (D) Box plot of the posterior/anterior pERK intensity ratio of wing discs of the designated genotypes. (E, F) Adult flies expressing the indicated transgenes under the control of *ap*-Gal4. Insets in E and F show confocal images of third instar wing imaginal discs expressing GFP and EGFRAP-GFP, respectively, under the control of *ap*-Gal4, stained with anti-GFP (green) and anti-EGFRAP (red). (G) Maximal projection of confocal images of an EGFR-sfGFP third instar wing imaginal disc expressing EGFRAP under the control of *Hh*-Gal4 stained with anti-GFP (green), anti-EGFRAP (red) and Hoechst (DNA, blue). (G'-G") Confocal *yz* sections, along the white dotted lines on the anterior (A) and posterior regions (P) of the wing disc shown in G. Apical side of the wing disc is to the left. The statistical significance of differences was assessed with a t-test, **** and ** P values <0.0001 and <0.01, respectively. Scale bars, 50 μm (A-C', G).

more rigorous control mechanisms to avoid the consequences of aberrant signaling (reviewed in [50]). Here, we have identified a novel EGFR inhibitor, *EGFRAP*, which exhibits a limited capacity to act as a negative regulator of EGFR during *Drosophila* wing morphogenesis, most likely due to redundancy with the SH2 domain containing protein PVRAP, yet behaves as an important regulator of the overgrowth driven by excessive activation of the EGFR/Ras pathway (Fig 8). We propose that the specificity of *EGFRAP* in cells with high levels of EGFR activity arises from activation of a negative feedback loop, which reduces excessive EGFR activity.

## The function of *EGFRAP* during morphogenesis and oncogenic transformation

The EGFR signaling pathway is a network with highly redundant and overlapping input signals and feedback controls (reviewed in [50]. This redundancy enhances robustness of the

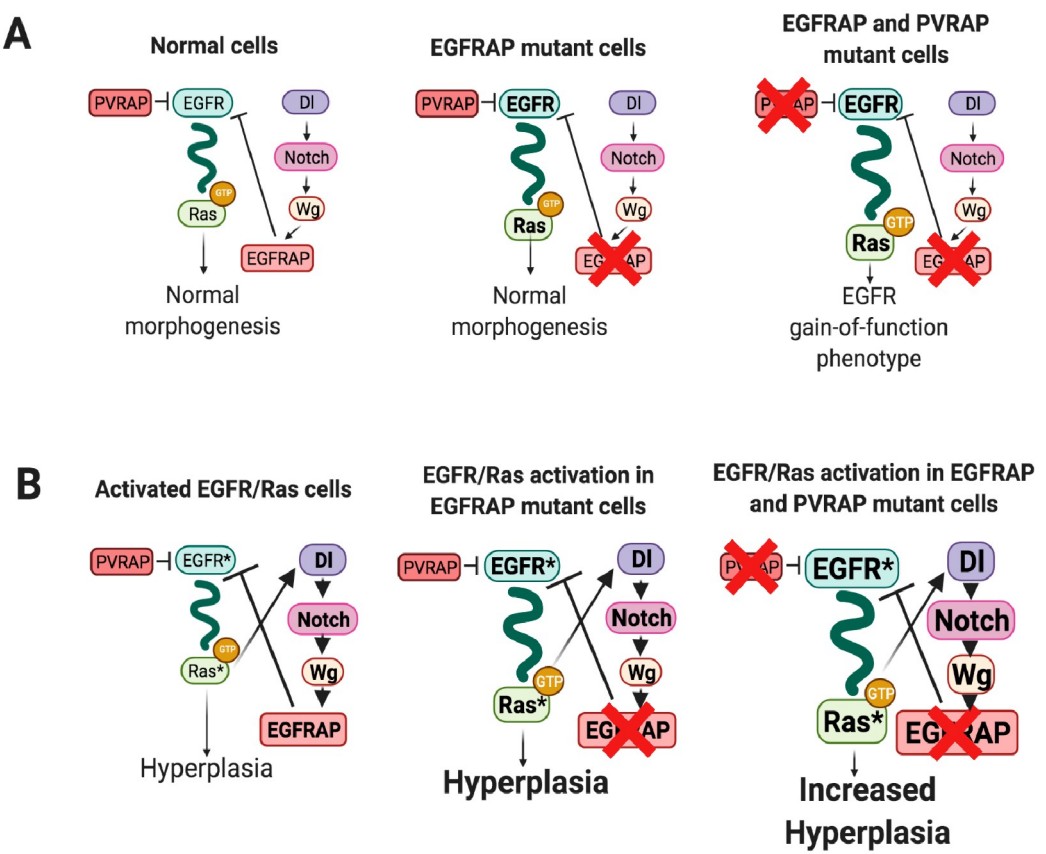

**Fig 8. Model of *EGFRAP* function as a modulator of EGFR/Ras-dependent tissue hyperplasia.** Schematic drawing depicting the mechanisms by which *EGFRAP* could limit EGFR/Ras activity in normal (A) and Ras^V12^-dependent oncogenic cells (B). (A) In normal cells *EGFRAP* and *PVRAP* act as negative regulator of the EGFR. Expression of *EGRAP* is confined to cells with high levels of EGFR activity via the Notch pathway. *EGFRAP* elimination results in a slight increase in EGFR activity, which does not seem to affect normal morphogenesis. However, the simultaneous elimination of *EGFRAP* and *PVRAP* leads to a further enhancement of EGFR signaling with consequences in normal morphogenesis. (B) Oncogenic EGFR/Ras activity promotes, via activation of the Notch pathway, an increase in *EGFRAP* expression that, in turn, restrains both EGFR activity and its capacity to induce hyperplasia. Downregulation of *EGFRAP* releases this restraint leading to a further increase in EGFR/Ras pathway activity and tumor growth, which is enhanced by elimination of *PVRAP*.

pathway, as dysfunction of a given regulator of the pathway can be compensated by others. The identification of *EGFRAP*, an SH2-domain containing protein that acts as a novel EGFR negative regulator, adds to the regulatory tool-box of the pathway. Elimination of *EGFRAP* in normal cells marginally disrupts EGFR pathway activity, without affecting overall morphogenesis or viability. However, downregulation of *EGFRAP* in Ras^V12^-expressing cells increases their oncogenic phenotype, suggesting that *EGFRAP* is needed to control high EGFR pathway activity. In this context, we report that next to *EGFRAP* lies *PVRAP*, a gene coding for another SH2 domain-containing protein, which we show that, similar to *PVRAP*, controls Ras^V12^-mediated tissue hyperplasia without affecting normal morphogenesis. PVRAP has been shown to physically interact with PVR [51], but based on the degree of homology of its SH2 domain with that of EGFRAP, we surmise it could also interact with EGFR. We therefore propose that EGFRAP and PVRAP might constitute a novel SH2 domain containing family of proteins that could function redundantly to regulate EGFR activity. In fact, the simultaneous

downregulation of *EGFRAP* and *PVRAP* in otherwise normal cells leads to defects consistent with EGFR hyperactivation, further emphasizing their regulatory role in the pathway. Additional biochemical, genetic and functional characterization of PVRAP will be required to confirm these findings and to gain a deeper understanding of how PVRAP interacts with EGFR in normal and pathological conditions.

Although, EGFRAP and PVRAP seem to work redundantly in normal cells, each of them can individually restrain EGFR/Ras signaling in the context of pathway hyperactivation. It is known that the main output of over-activated EGFR/Ras signaling is the induction of a complex and dynamic set of transcriptional networks, leading to changes in the gene expression signature and metabolic state of cancer cells [52]. As a result of these changes, new genetic interactions can arise in tumor cells [53]. In this context, genes with non-essential or redundant functions in normal cells can become highly relevant in tumorigenic conditions. For example, expression of the copper/zinc dismutase SOD1 is dispensable for normal mammary gland development, yet is essential for the survival of breast cancer cells. SOD1 is specifically required in cancer cells to counter the increased superoxide production associated with oncogene activation [54]. Similarly, the MTH1 protein, which sanitizes oxidized dNTP pools, is required for cancer cell survival, but is dispensable for untransformed cells viability. The requirement of MTH1 in cancer cells arises from their need to adapt to an imbalanced redox state not seen in normal cells [55]. *Drosophila Socs36E* and its human ortholog, SOCS5, have a limited capacity to act as growth regulators under normal conditions but have proved to be important regulators of EGFR-dependent cellular transformation [56–58]. Likewise, FAK moderates RTK signaling only in situations of pathway hyper-activation [59]. Interestingly, in all these cases, oncogene activation promotes increased expression of the otherwise non-essential genes by mechanisms that remain poorly understood. Similarly, we show here that *EGFRAP* expression is upregulated by EGFR overactivation.

Based on the above and on our results, we propose that the ability of *EGFRAP* to functionally inhibit EGFR/Ras activity in Ras$^{V12}$-expressing cells might arise from its increased levels of expression in these oncogenic cells. The fact that overexpression of *EGFRAP* in normal cells results in phenotypes associated with EGFR loss-of-function supports this view. Finally, we find that the Notch pathway, which is itself upregulated by hyperactivated EGFR/Ras signaling [60], is required for the increase in EGFRAP expression due to EGFR/Ras hyperactivation. Thus, we propose a model by which cells minimize the effect of excessive EGFR/Ras signaling by inducing the expression of negative regulators of the EGFR/Ras pathway, such as EGFRAP, via Notch activation.

EGFR autocrine loop is required for transformation by activated Ras in different mammalian cell systems, including fibroblasts, keratinocytes, intestinal epithelial cells, melanomas and pancreatic cells [21–23]. The contribution of EGFR to oncogenic transformation seems to be evolutionary conserved, as it has also been found in *Drosophila* [30] and in *Xiphophorus* fish [61]. Furthermore, EGFR stimulates Ras-dependent tumor overgrowth through canonical EGFR signaling. Thus, even though oncogenic Ras isoforms are constitutively active, EGFR activation can induce GTP loading on wild type Ras isoforms, enhancing effector pathway signaling [62]. In this context, recent studies in mice indicate the existence of an optimal level of Ras activation, the so-called "sweet spot", conducive to tumor formation and progression (reviewed in [63]. Thus, any negative regulator of the EGFR, such as *EGFRAP* and *PVRAP*, could act as tumor suppressors of the overgrowth of wild type cells ectopically expressing Ras$^{V12}$. Elimination of *EGFRAP* and/or *PVRAP* in these cells would lead to hyperactivation of the EGFR, which, in turn, would result in an enhancement of endogenous wild-type Ras signaling, thus increasing pathway activity and overgrowth. Alternatively, or in addition, the increase in EGFR activity due to elimination of *EGFRAP* and/or *PVRAP* could induce further

overgrowth of Ras$^{V12}$ cells by stimulating non-canonical EGFR pathways. In fact, the EGFR stimulates the growth of fly and human cancer cells harboring oncogenic Ras mutations through the activation of pathways controlling cell growth and survival, such as the Hedgehog pathway [24].

## *EGFRAP* enhances Ras$^{V12}$-dependent tissue hyperplasia

Here, we find that the most prevalent change in wing disc morphology caused by Ras$^{V12}$ over-expression is the formation of extra folds, a phenotype that increases upon the removal of *EGFRAP*. We also show that the appearance of ectopic folds is associated with changes in cell shape, from columnar to cuboidal, and to a rise in cell volume. Increased EGFR signaling has been shown to affect cell shape through the regulation of Myosin II dynamics and FAK inactivation [64,65]. Thus, the increased cell shape changes observed in Ras$^{V12}$ cells upon *EGFRAP* down-regulation could be a direct consequence of the role of *EGFRAP* as a negative regulator of EGFR activity.

The EGFRAP SH2 domain most closely resembles those of the Tensins (FlyBase). Tensins are a family of focal adhesion proteins, composed of four members (Tensin 1–4, TNS1-4), which link the cell membrane to the actin cytoskeleton and are lost in most cancer cell lines [66,67]. EGFRAP and PVRAP have been proposed to be orthologs of human TNS2 and TNS4 (FlyBase), proteins that in knockdown conditions increase tumorigenicity in several cancer lines [68]. In addition, TNS4 levels increase following EGF stimulation [69], in the same way as we observed for EGFRAP. Furthermore, TNS4 regulates cell shape downstream of EGFR, via interaction through their SH2 domains [69]. Finally, the role of Tensins as tumor suppressors has also been linked to their ability to bind and regulate integrins (reviewed in [66]. In this work, we observed that downregulation of both *EGFRAP* and PVRAP results in wing blisters, a phenotype associated with the loss of integrin function [70]. In this context, we propose that *EGFRAP*, and possibly PVRAP, could regulate cell shape changes downstream of EGFR hyperactivation in a similar way to the mammalian tensins.

## *EGFRAP* as a negative regulator of EGFR activity

Negative feedback loops are present at multiple levels of the EGFR signaling pathway including ligand sequestration, direct association with EGFR, blockade of Ras binding to receptor complexes, and the activation of MAPK or downstream transcription factors [71–75]. It is within this framework of extensive negative feedback regulation where the function of *EGFRAP* can be better understood. Our two-hybrid analysis showed a strong specificity of EGFRAP towards phosphorylated EGFR, suggesting that receptor activation is a prerequisite for EGFRAP binding to EGFR. EGFRAP binding to the EGFR depends on the phosphorylation of ten specific tyrosine residues located in its cytoplasmic domain. Recent experiments have demonstrated that each set of phosphorylated tyrosines can lead to the recruitment of a unique set of downstream factors, eliciting distinct signaling durations and levels [76]. Diverse EGFR signaling intensities can then trigger different responses. For example, low levels of EGFR activity are needed for wing imaginal cell survival and proliferation, whereas high levels of signaling are required to initiate wing vein formation [77]. In this context, EGFRAP could act as a weak competitor of EGFR downstream effectors, binding the same EGFR tyrosines as EGFRAP. Alternatively, or in addition, EGFRAP could act as a weak competitor of Shc, as it binds weakly to the same site to which Shc binds, Y1357 [40]. Both scenarios would explain why the elimination of EGFRAP does not result in morphological defects, while its overexpression, which occurs in conditions of pathway hyperactivation, strongly inhibits EGFR function.

Endocytic trafficking has a central role in the regulation of EGFR signaling. After internalization, EGFR can be recycled to the cell membrane or targeted for degradation. Internalized receptors targeted for lysosomal degradation require the family of Suppressor of Cytokine Signaling (SOCS), among other factors [78,79]. Similar to EGFRAP, SOCS proteins, such as *Drosophila* Socs36E and its human ortholog SOCS5, also contain an SH2 domain, are induced by EGF stimulation and behave as tumor suppressors in cooperation with the EGFR/Ras pathway [56–58]. However, here we show that the levels and localization of EGFR in wing margin cells are not affected by altered *EGFRAP* expression. These results suggest that *EGFRAP* might regulate EGFR activity independently of its trafficking. Deciphering the ways in which EGFRAP regulates EGFR activity certainly deserves further investigation.

Here, we have identified a new EGFR inhibitor, *EGFRAP*, whose function during normal morphogenesis seems to be redundant, yet behaves as a significant regulator of cellular overgrowth driven by excessive EGFR/Ras signaling. The identification of genetic mechanisms to be deployed mainly in tumorigenic contexts could help in the development of new therapeutic drugs targeting cancer but not normal cells, a top priority in cancer research.

## Materials and methods

### Fly strains

The following stocks were used: *UAS-EGFRAP^RNAi^*, *UAS-PVRAP^RNAi^* (Vienna Stock Center); UAS-Ras^V12^ [80]; UAS-λtop; UAS-EGFR^DN^; UAS-N^i^; UAS-N RNAi; *UAS-Diap1, en-Gal4-UAS-mRFP/CyO, ap-Gal4; Hh-LacZ* (Bloomington Drosophila Stock Center); ap>myrT (agift from M. Milan); *tub-Gal80^ts^; ptc-Gal4* (ptc^80ts^Gal4, a gift from Dra. Isabel Guerrero), *hh-Gal4-UAS-GFP/TM6b* (a gift from Prof. F. Casares), EGFR-sfGFP (a gift from Dr. N. Yakobi, [39] and *nub-Gal4* [81]. The mutants KD, L/S and L were generated by CRISPR in this study (see below). Flies were raised at 25˚ C. For RNAi experiments using the ptc^80ts^Gal4 line, crosses were grown at 18˚ C for 6 days, followed by transfer to 29˚ C and dissection two days later (L3 stage).

### Transgenic flies

The *UAS*-EGFRAP construct was made using an EcoR1+Not1 fragment from EST RE08107 cloned in pUASt. The *UAS*-EGFRAP-*GFP* construct was made using two *EcoRI+PstI* and *PstI +SacII* fragments from EST RE08107, which were cloned into the *EcoRI* and *SacII* sites of the pEGFP plasmid. From this, an *EcoRI+NotI* fragment was purified and cloned into the pUAST vector. The resulting construct was introduced into the germ-line following standard procedures to produce several transgenic lines.

### Generation of *EGFRAP* mutants with CRISPR/Cas9

SgRNAs were designed to generate three different mutants: sgRNA1 was designed against sequences located in the 3rd exon to generate mutants eliminating the two isoforms encoded by *EGFRAP*, *EGFRAP^L/S^*; sgRNA2 was designed against sequences located in the 1st exon to generate mutants eliminating only the long isoform, *EGFRAP^L^*, and sgRNA3 was designed against sequences located at the beginning of the 5th exon to generate mutants lacking the SH2 domain, *EGFRAP^ΔSH2^*. We used the following sequences:

sgRNA1: `GTCGAGTGCCACCCAGCGGCTGC`

sgRNA2: `GTCGCTCCGATGGCTATGTGAACG`

sgRNA3: `GTCGTGTGCTGTGGTTCCAGGCGT`

sgRNA4: GTCGACAGCTCCTGGACATCGTGG

sgRNA5: GTCGTTGCGATCGGGTACTAGGT

sgRNA6: GTCGCTGATCCTGCGGACGCAAAG

The sgRNAs were cloned into the PCFD3 vector as previously described [82] and http://www.crisprflydesign.org/plasmids/. Transgenic gRNA flies were generated by the Best Gene Company (Chino Hills, USA) using either *y sc v P{nos-phiC31\int.NLS}X; P{CaryP}attP2* (BDSC 25710) or *y v P{nos-phiC31\int.NLS}X; P{CaryP}attP40* (BDSC 25709) flies. Transgenic lines were verified by sequencing by Biomedal (Armilla, Spain). Males carrying the sgRNA were crossed to nos-Cas9 females and the progeny was screened for the v+ch- eye marker. To identify CRISPR/Cas9-induced mutations, genomic DNA was isolated from flies and sequenced using the following primers (5'-3'):

*EGFRAP* primer sgRNA1 Forward: ACATGCAACATGCAAATCGT

*EGFRAP* primer sgRNA1 Reverse: GGAAACAATTTCGGACTGGA

*EGFRAP* primer sgRNA2 Forward: ACATGCAACATGCAAATCGT

*EGFRAP* primer sgRNA2 Reverse: GGAAACAATTTCGGACTGGA

*EGFRAP* primer sgRNA3 Forward: ACACGTTTTTGAACAGATGCTT

*EGFRAP* primer sgRNA3 Reverse: GCGACTTCTCAGCTGTCTCA

*EGFRAP* primer sgRNA4 Forward: AGCTGCTGGAACAGTGACAA

*EGFRAP* primer sgRNA4 Reverse: CCCTCGCAGAACTTACCAGC

*EGFRAP* primer sgRNA5 Forward: CAGCGAGCCATACATCTCGAA

*EGFRAP* primer sgRNA5 Reverse: ACTCGGTGATGCCGGAACT

*EGFRAP* primer sgRNA6 Forward: GTGCTCGGACTTACCTTTATCT

*EGFRAP* primer sgRNA6 Reverse: GATGTTTCGCCCTCTCGCT

Two *EGFRAP* mutant alleles were generated, in which both isoforms (*EGFRAP$^{L/S}$*) or only the long isoform (*EGFRAP$^L$*) were eliminated (Fig 4A). Both alleles were deletions of 7 and 23 base pairs, respectively, which resulted in frame-shifts generating stop codons 14 and 26 amino acids after the shift. In addition, a third allele was generated carrying a deletion of 10 base pairs and a stop codon 3 amino acids after the shift, which resulted in the complete elimination of the SH2 domain (Fig 4A).

## Immunocytochemistry, *in situ* hybridization, adult wing mounting and imaging

Wing imaginal discs were stained using standard procedures and mounted in Vectashield (Vector Laboratories, Burlingame, California). The following primary antibodies were used: goat anti-GFP$^{FICT}$ (Abcam, 1:500), rabbit anti-PKC (Santa Cruz Biotechnology, 1:300), mouse anti-Dlg (DSHB, 1:50), mouse anti-Elav (1/500, DSHB), rabbit anti-PHH3 (EMD Millipore Corporation; 1:250), rabbit anti-caspase Dcp1 (Cell Signaling; 1:100), rabbit anti-pJNK (Promega, 1:200), mouse anti-βGal (Promega, 1:1000), rat anti-Sens (DSHB, 1:100), rabbit anti-EGFRAP (this study, 1:100–1:200), rabbit anti-dpERK (Cell Signaling, 1:20) and sheep α-DIG AP (Roche, 1:2000). The secondary antibodies used were: goat anti-mouse Alexa-488, Cy3 and Cy5 (Life Technologies, 1:200) goat anti-rabbit Cy3 and Cy5 (Life Technologies, 1:200) and

goat anti-rat Cy3 (Life Technologies, 1:200). F-actin was visualized using Rhodamine Phalloidin (Molecular Probes, 1:50). DNA was marked using Hoechst (Molecular Probes, 1:1000).

Confocal images were obtained using a Leica SP5-MP-AOBS or a Zeiss LSM 880 microscope, equipped with a Plan-Apochromat 63Xoil objective (NA 1.4).

*In situ* hybridization was performed using standard procedures. A digoxygenin-UTP (Boerhringer-Mannheim) labelled *EGFRAP* anti-sense RNA probe was generated using the plasmid cDNA EST RE08107 (Drosophila Genomics Resource Center).

Adult wings were extracted in 70% ethanol, mounted onto slides in Hoyer's medium and imaged with a Nikon camera attached to a Zeiss Axioplan2 microscope.

## Quantification of fluorescence intensity

For quantification of fluorescent intensity of different markers, fluorescent signaling was measured on several confocal images per genotype using the line tool in FIJI-Image J.

For calculation of cell areas, the Huang threshold algorithm was applied to maximum projections of confocal sections. Cell volumes were calculated considering wing disc cells as truncated prisms and applying the formula Volume = Height/3 (Basal Area + Apical Area + $\sqrt{}$ Basal Area x Apical Area).

For the quantification of cell death, the Trainable Weka 2D Segmentation plug-in, which transforms 8-bit images into a binary system, was used to measure fluorescent intensity of wing discs stained with an anti-Dcp1 antibody.

For quantification of dp-ERK levels in wing discs, the region of interest was selected manually using the FIJI-ImageJ tracing tool. Measurements represented in the graph correspond to the average mean dp-ERK intensity ratios between the posterior experimental versus the anterior control regions.

We used Student's t tests for statistical comparisons of fluorescence intensity values.

## Generation of the anti-EGFRAP antibody

The coding sequence corresponding to aa 163 to 274 of the EGFRAP N-terminal region was amplified by PCR. This region is common to both forms and excludes the SH2 domain, which is conserved between many other *D. melanogaster* proteins. To increase translation efficiency, a codon optimization tool was used (IDT company). The PCR product was inserted into the plasmid pXTB1, which contains the sequence for an intein tag. Peptide purification was performed using an intein-chitin affinity column (IMPACT). The target peptide was further purified by high-performance liquid chromatography. Rabbits were immunized using the purified peptide. The anti-EGFRAP antibody was purified from the original serum using peptide bound to an intein-chitin affinity column. Specificity of the anti-EGFRAP was confirmed by Western Blot.

## Yeast two hybrid

Two hybrid assays and library screening were done according to standard procedures, as previously described [40,83,84]. Briefly, we used a cDNA Library from *Drosophila* third instar larvae made using the pSE1107 plasmid, generously provided by Dr Grace Gill (Harvard Medical School, Boston, MA). Positive interactions were selected by plating the transformation on restrictive medium with double nutritional requirement; growing colonies were re-streaked and further tested for β-galactosidase activity; DNA was recovered from the positive clones and sequenced. The library was screened using the wild-type PDGF and VEGF-receptor Related (PVR) intracellular domain (PVRi) fused to Gal4 DNA binding domain as a bait.

A battery of baits was then used to determine the interaction specificity, to map these interactions to specific residues and to test kinase activity requirement. These baits included several Epidermal Growth Factor Receptor (EGFR) intracellular domain (EGFRi) baits, either with the wild-type sequence or with specific tyrosine residues mutated to phenylalanines. Generation of EGFR baits and point mutants has been described in detail in [40,84]. A Fibroblast Growth Factor Receptor 2/breathless (FGFR2/btl) intracellular domain bait and a Fibroblast Growth Factor Receptor 1/heartless (FGFR1/btl) intracellular domain bait were also used to determine interaction specificity. Both were constructed by PCR, using specific primers and either λ-htl [85] or λ-btl [86] as templates. Amplified inserts were then cloned into pGBKT7 bait plasmid (Takara Bio) and verified by sequencing. Binding of the mouse p53 protein to SV40 large T-antigen or to PVR-IP#102 were used as positive or negative controls, respectively.

## Supporting information

**S1 Fig. Reducing *EGFRAP* levels in a narrow region of the wing disc increases Ras<sup>V12</sup>-dependent tissue hyperplasia.** (A-C') Maximal projection of confocal images of wing imaginal discs from third-instar larvae expressing the indicated UAS transgenes under the control of $ptc^{80ts}$-Gal4 line stained with anti-GFP (green) and RhPh (red). (D) Box plot of the area of the wing disc occupied by GFP+ cells of the indicated genotypes. The statistical significance of differences was assessed with a t-test, ****P value<0.0001. Scale bars, 50 μm (A-C). (TIF)

**S2 Fig. Downregulation of *EGFRAP* does not affect the proliferation of Ras<sup>V12</sup> imaginal disc cells.** (A-C') Maximal projection of confocal images of wing imaginal discs from third-instar larvae expressing the indicated UAS transgenes under the control of *ap*-Gal4, stained with anti-GFP (green), anti-Phosphohistone 3 (PH3, red) and Hoechst (DNA, blue). (D) Box plots of total number of PH3+ cells/disc of the designated genotypes. The statistical significance of differences was assessed with a t-test, **** and *** P values <0.0001 and <0.001, respectively. Scale bars, 50 μm (A-C). (TIF)

**S3 Fig. Reducing *EGFRAP* levels in Ras<sup>V12</sup> wing discs increases the ability of tumor cells to migrate into surrounding wild-type tissue.** (A-C') Maximal projection of confocal images of wing imaginal discs from third-instar larvae expressing the indicated UAS transgenes under the control of *ap*-Gal4, stained with anti-GFP (green) and RhPh (red). (D-D") Confocal cross-sections of an *ap*>GFP; Ras<sup>V12</sup>; *EGFRAP*<sup>RNAi</sup> wing disc stained anti-GFP (green), anti-Dcp1 (red) and Hoechst (DNA, blue). White arrows point to dorsal tumor cells (GFP+) invading the ventral compartment. White arrowheads point to control ventral cells (GFP-) undergoing apoptosis. (E) Quantification of wing discs of the indicated genotypes with (pale grey) or without (strong grey) dorsal tumor cells (GFP<sup>+</sup>) in the ventral compartment. Scale bars, 50 μm (A-C) and 30 μm (D-D"). (TIF)

**S4 Fig. Downregulation of *EGFRAP* in Ras<sup>V12</sup> tumor cells does not alter the distribution of polarity markers, such as Dlg and aPKC.** (A-F) Maximal projection of confocal views of third instar wing imaginal discs expressing the indicated UAS transgenes under the control of *ap*-Gal4, stained with anti-GFP (green), anti-aPKC (A-C, red) or anti-Dlg (D-F, red) and Hoechst (DNA, blue). (A'-A"-F'-F") Confocal sections of wing discs of the indicated genotypes along the white dotted lines shown in A-F, respectively, parallel (A'-F') or perpendicular (A"-F") to the A/P axis. Apical sides of wing discs are to the left (A'-F') or top (A"-F"). Scale bars,

50 μm (A-F).
(TIF)

**S5 Fig.** *EGFRAP*[L] and *EGFRAP*[ΔSH2] **behave as** *EGFRAP*[L/S] **in their ability to synergize with** *Ras*[V12] **to drive hyperplasia in wing imaginal discs.** (A-C) Maximal projection of confocal views of third instar wing imaginal discs of the indicated genotypes, stained with anti-GFP (green), RhPh (red) and Hoechst (DNA, blue). (A'-A"-C'-C") Confocal sections of wing discs of the indicated genotypes along the white lines shown in A-C, respectively, parallel (A', B' and C') or perpendicular (A", B", and C") to the A/P axis. Apical sides of wing discs are to the left (A', B' and C') or top (A", B" and C"). (D) *EGFRAP*[L/S] mutant wing disc stained with anti-*EGFRAP*. (E) Wing disc expressing a *EGFRAP* RNAi with *en-Gal4*, stained with anti-*EGFRAP* (red) and Hoechst (DNA, blue). Scale bars, 50 μm (A-E).
(TIF)

**S6 Fig. Direct binding of EGFRAP to EGFR.** (A) Scheme of EGFR, the different baits used in the yeast two-hybrid assays and the tyrosine to phenylalanine mutations introduced in the cytoplasmic domain of EGFR (EGFRi). (B) Left: scheme of the EGFRAP-PA protein, and probable length of the partial clone PVR-IP#102 (black bar). Right: Interactions detected in yeast cells co-transformed with the PVR-IP#102 prey and different baits. -WL, medium lacking Trp and Leu. -WL+Xgal, medium lacking Trp and Leu supplemented with Xgal substrate. -WLHA, medium lacking Trp, Leu, His and Adenine. Interactions between candidate proteins were assessed by analyzing both β-galactosidase activity (-WL+Xgal) and the ability to grow in the absence of His and Adenine (-WLHA): ++ very strong interaction; + strong interaction: w, weak interaction; vw, very weak interaction;—no interaction; relative to the positive control. C +, positive control; C-, negative control.
(TIF)

**S7 Fig. Expression and requirement of EGFRAP in** *Drosophila* **eye imaginal discs.** (A-D') Confocal views of wild-type (A-B' and D, D') and Hedgehog-LacZ (Hh-LZ, C, C') eye imaginal discs stained with anti-Elav (green in and B, white in A'), anti-EGFRAP (red in A, C, C', D and D' and white in A" and B'), anti-βgal (green in C, C'), anti-Senseless (Sens, green in D, D') and Hoechst (DNA, blue, A, B, C, D and E). B-B', C' and D' are magnifications of the white boxes in A, C and D, respectively. (E, E') Eye discs carrying small clones of λtop expressing cells (GFP+) stained with anti-GFP (green), anti-EGFRAP (red in E, white in E') and Hoechst (DNA, blue). (F, G) Adult female *Drosophila* eyes of the indicated genotypes. Scale bars, 50 μm (A-A", C, D); 10 μm (B, B', C', D'), 5 μm (E, E').
(TIF)

**S8 Fig.** *EGFRAP* **Knock down enhances λtop-associated tissue hyperplasia in** *Drosophila* **wing imaginal discs.** (A-C') Maximal projection of confocal views of wing imaginal discs from third-instar larvae expressing the indicated UAS transgenes under the control of *ptc*[80ts]-Gal4 stained with anti-GFP (green), RhPh (red) and Hoechst (DNA, blue). Scale bars, 50 μm (A-C).
(TIF)

**S1 Data. Numerical Data underlying graph Fig 2J.** File containing numerical raw data corresponding to Fig 2J.
(JPG)

**S2 Data. Numerical Data underlying graph Fig 2K.** File containing numerical raw data corresponding to Fig 2K.
(JPG)

**S3 Data. Numerical Data underlying graph Fig 2L.** File containing numerical raw data corresponding to Fig 2L.
(JPG)

**S4 Data. Numerical Data underlying graph Fig 3D.** File containing numerical raw data corresponding to Fig 3D.
(JPG)

**S5 Data. Numerical Data underlying graph Fig 3H.** File containing numerical raw data corresponding to Fig 3H.
(JPG)

**S6 Data. Numerical Data underlying graph Fig 6C.** File containing numerical raw data corresponding to Fig 6C.
(JPG)

**S7 Data. Numerical Data underlying graph Fig 6J.** File containing numerical raw data corresponding to Fig 6J.
(JPG)

**S8 Data. Numerical Data underlying graph Fig 7D.** File containing numerical raw data corresponding to Fig 7D.
(JPG)

**S9 Data. Numerical Data underlying graph S1D Fig.** File containing numerical raw data corresponding to S1D Fig.
(JPG)

**S10 Data. Numerical Data underlying graph S2D Fig.** File containing numerical raw data corresponding to S2D Fig.
(JPG)

**S11 Data. Numerical Data underlying graph S3E Fig.** File containing numerical raw data corresponding to S3E Fig.
(JPG)

## Acknowledgments

We thank Bloomington and Kyoto Stock Centers and the *Drosophila* community for fly stocks and reagents. We also thank Dr. John R. Pearson for proofreading corrections. CML thanks Prof. I. Correas (Universidad Autónoma de Madrid, Spain) and Dr M.A. Alonso (CSIC, Spain) for their constant support and generosity.

## Author Contributions

**Conceptualization:** Jennifer Soler Beatty, Cristina Molnar, Jose F. de Celis, María D. Martín-Bermudo.

**Formal analysis:** Jennifer Soler Beatty, Cristina Molnar, Carlos M. Luque, Jose F. de Celis, María D. Martín-Bermudo.

**Funding acquisition:** Carlos M. Luque, Jose F. de Celis, María D. Martín-Bermudo.

**Investigation:** Jennifer Soler Beatty, Cristina Molnar, Carlos M. Luque, Jose F. de Celis, María D. Martín-Bermudo.

**Methodology:** Jennifer Soler Beatty, Cristina Molnar, Carlos M. Luque, Jose F. de Celis, María D. Martín-Bermudo.

**Project administration:** María D. Martín-Bermudo.

**Resources:** Jose F. de Celis, María D. Martín-Bermudo.

**Supervision:** Jose F. de Celis, María D. Martín-Bermudo.

**Validation:** María D. Martín-Bermudo.

**Writing – original draft:** Jennifer Soler Beatty, María D. Martín-Bermudo.

**Writing – review & editing:** Jennifer Soler Beatty, Cristina Molnar, Carlos M. Luque, Jose F. de Celis, María D. Martín-Bermudo.

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
