## [Decision Letter · Decision Letter 0]

8 Mar 2021

Dear Dr Martin-Bermudo,

Thank you very much for submitting your Research Article entitled 'CG33993 encodes a new SH2 domain-containing protein acting as a negative feedback loop of EGFR/Ras-driven tissue hyperplasia' to PLOS Genetics.

The manuscript was fully evaluated at the editorial level and by independent peer reviewers. The reviewers appreciated the attention to an important problem, but raised some substantial concerns about the current manuscript. Based on the reviews, we will not be able to accept this version of the manuscript, but we would be willing to review a much-revised version. We cannot, of course, promise publication at that time.

Should you decide to revise the manuscript for further consideration here, your revisions should address all the specific points made by each reviewer. We expect the revised manuscript will include new data on the functional relationships between CG33993 and EGFR signaling in normal developing tissues, which will address the concern whether CG33993 is a true negative regulator of the EGFR/Ras pathway that is induced by EGFR signaling. We will also require a detailed list of your responses to the review comments and a description of the changes you have made in the manuscript.

If you decide to revise the manuscript for further consideration at PLOS Genetics, please aim to resubmit within the next 60 days, unless it will take extra time to address the concerns of the reviewers, in which case we would appreciate an expected resubmission date by email to plosgenetics@plos.org.

[LINK]

We are sorry that we cannot be more positive about your manuscript at this stage. Please do not hesitate to contact us if you have any concerns or questions.

Yours sincerely,

Wei Du

Guest Editor

PLOS Genetics

Gregory P. Copenhaver

Editor-in-Chief

PLOS Genetics

Reviewer's Responses to Questions

**Comments to the Authors:**

Reviewer #1: The paper by Soler Beatty describes a screen for inhibitors of activated EGFR/Ras signaling. Expression of activated Ras in the wing disc leads to deformed discs, and this phenotype was exacerbated by co-expression of CG33993 RNAi. The presence of an SH2 domain in CG33993 indicated a possible role in signaling. CG33993 null mutants display no phenotype. Normal expression of CG33993 is observed in some regions of EGFR activity but not in many others, and is upregulated following constitutive EGFR/Ras activity. Finally, over expression of CG33993 attenuates EGFR signaling.

Evaluating this paper is a close call, because the biological significance of CG33993 is unclear. Most of the conclusions were obtained under ectopic conditions. CG33993 RNAi phenotype was revealed only following expression of activated EGFR/Ras, while null mutants have no phenotype on their own. The normal expression pattern coincides only marginally with regions of prominent EGFR activation. It is however upregulated following constitutive EGFR/Ras activity, indicating some connection. The phenotype of over-expressed CG33993 could represent an indirect dominant-negative effect. Thus, it is clear that CG33993 has the capacity to attenuate EGFR/Ras signaling, possibly due to dominant-negative effects of its SH2 domain. However, whether it normally plays any role as an inducible inhibitor of the pathway is not convincingly demonstrated. I would recommend examining the normal expression pattern of CG33993 and expression after ectopic EGFR activation in additional tissues, such as the embryo or the eye disc, to solidify these findings which are the strongest aspect of the paper.

In the model (Fig. 8), it is not clear to me why the authors suggest that enhanced signaling downstream of the receptor, in the absence of CG33993, will lead to excess ligand processing. Rhomboid, the Spitz protease, is a downstream target of the pathway only in a very restricted set of tissues (eye disc and ovary).

Finally, I did not find the Figure Legends section.

Reviewer #2: In this paper, Beatty et al. identify the SH2-domain protein CG33993 as a novel feedback inhibitor of EGFR signaling. They show that knocking down CG33993 by RNAi enhances the wing disc overgrowth induced by expression of activated Ras. The effect seems to be largely due to increased cell growth and cell shape changes. Removing CG33993 or its SH2 domain by mutation has the same enhancing effect, but causes no phenotype in an otherwise wild-type background. CG33993 interacts with the intracellular domain of EGFR in a yeast two-hybrid assay, and mutating a series of tyrosines in EGFR diminishes this interaction. CG33993 is expressed adjacent to the DV boundary of the wing disc, in a domain of high EGFR activity, and the protein localizes apically within the cells. Increasing CG33993 levels reduces dpERK levels and alters wing morphology, and CG33993 levels are increased by EGFR activation.

These results are potentially interesting, as CG33993 homologues could act as tumor suppressors (although the authors do not discuss whether clear human homologues exist). However, the paper could be improved by addressing some additional points.

1) The authors do not show that normal CG33993 expression requires EGFR signaling. How is it affected in cells mutant for EGFR or a downstream component of the pathway? Why is it only expressed at the wing margin and not in the wing vein primordia?

2) The authors do not provide a mechanism by which CG33993 binding to the EGFR inhibits EGFR function. Is the level of EGFR protein or its intracellular localization altered by CG33993 overexpression?

3) It is not clear that the intracellular tyrosines in EGFR would undergo the appropriate phosphorylation in yeast. Can the authors show that mutating these to acidic residues maintains or increases the interaction, as predicted if it depends on phosphorylation?

Minor points:

1) A control disc with ap-GAL4 driving only CG33993 RNAi should be included in Fig. 1.

2) In Fig. S2, PH3 would be better quantified as the number of labeled cells per disc rather than mean fluorescence intensity.

3) It would much improve readability if the authors used the name they have chosen for their gene, loef, rather than CG33993, throughout the manuscript.

4) The paper should be carefully proof-read. In addition to the lack of figure legends in the original submission, there are errors throughout. For instance, the numbers in the text do not match those in the graphs in Fig. 2K, L, it is unclear from the text and Fig. 5 whether CG33993 interacts with Btl or Htl, and when describing the generation of the CG33993 antibody, the authors state that region X-Y of the coding sequence was amplified by PCR.

Reviewer #3: The manuscript describes characterization of CG33993, a Drosophila gene encoding a protein with an SH2 domain. Mutation or knockdown of CG33993 had no discernible phenotype in normal development. However, the gene was discovered in an RNAi screen as an enhancer of RasV12 phenotypes, suggesting the potential to inhibit pathway activity. A detailed examination in the wing disc showed RasV12 hyperplasia was enhanced by loss of function of CG33993 to exacerbate cell shape changes. CG33993 appears to be a target of the pathway and was induced when constitutively active EGFR was ectopically expressed. The connection to the EGFR pathway was further established through yeast-two-hybrid assays that showed interaction of CG33993 with certain residues in the EGFR intercellular domain known to interact with SH2 domain proteins. The authors conclude CG33993 participates in a negative feedback loop and suggest it would have the potential to modify EGFR/Ras activity in cases of elevated signaling, even though a role in normal physiology was not observed.

The work is of a high standard and represents a thorough characterization of the gene including expression analysis to the level of the sub-cellular localization of the protein. Genetic evidence that CG33993 can inhibit EGFR/Ras is convincing. The yeast-two-hybrid results are a good starting point towards defining the molecular mechanism whereby CG33993 could bind and inhibit EGFR. However, the description of the various mutants was quite confusing. The details could be confined to the figure and the take-home lesson kept in the text. Given that an antibody is in hand for both proteins (tagged reagents could also be used), it should be possible to demonstrate this interaction is physiological by IP experiments in Drosophila cells and this would provide stronger evidence.

The idea that genetic mechanisms exist only to be deployed in aberrant signaling situations is an interesting one and the authors cite some examples. However, the overall case of the importance of studying CG33993 to understand the role of Ras in tumorigenesis fell short of being convincing. A more robust discussion of why non-essential genes are interesting and citing more relevant literature could bolster the case. The closing remarks about finding more genes like CG33993 for their value as therapeutic targets in cancer was weak. Rather than increasing enthusiasm for the project, this emphasized that this detailed study of CG33993 had not cast much light on its own potential in this regard.

Related to the above point, redundancy could be masking a phenotype in the mutant. Very little discussion of the molecular structure of the CG33993 was provided, but there was some mention of the related Tensin1-3 genes. Are there any genetic interactions with these genes? Moreover, the authors cite the correlation of Tensin1-3 loss in cancer, which prompts the question of what about CG33993? I think it may not be easy to recognize mammalian homologs if there are no domains in addition to the SH2 domain, but a fuller discussion of this is needed, or the reader is left with questions about the molecular nature of the gene and whether it could be relevant to cancer.

The manuscript needs attention. For example, there is a repeated sentence in the abstract, references are not fully formatted, there were no figure legends (that I could find) and throughout the text would benefit from consultation with a copy editor. Reducing the length of the manuscript (by as much as half) would improve impact and readability.

Small point: The authors suggest a name for the gene but do not use it.

**Have all data underlying the figures and results presented in the manuscript been provided?**

Reviewer #1: Yes

Reviewer #2: **No: **There are graphs in several of the figures for which spreadsheets are not provided.

Reviewer #3: **No: **This may not be essential, but I wondered about the RNAi screen that resulted in the present study of CG33993. A supplemental figure with these details would be useful. Especially given the lack of any phenotype except in sensitized backgrounds--were other genes of interest discovered and why given that proviso was CG33993 chosen for detailed analysis?

PLOS authors have the option to publish the peer review history of their article (what does this mean?). If published, this will include your full peer review and any attached files.

Reviewer #1: No

Reviewer #2: No

Reviewer #3: No

---

## [Decision Letter · Decision Letter 1]

21 Jun 2021

Dear Dr Martin-Bermudo,

Thank you very much for submitting your Research Article entitled 'EGFRAP encodes a new SH2 domain-containing protein acting in a negative feedback loop that controls EGFR/Ras-driven tissue hyperplasia' to PLOS Genetics.

The manuscript was fully evaluated at the editorial level and by independent peer reviewers. While the revised manuscript is significantly improved, the reviewers still have significant concerns, particularly with regard to the presentation/discussion of results in light of the new findings and with regard to demonstrating the binding solely based on yeast two-hybrid assay. In addition, since Notch signaling is highly activated in the wing margin while EGFRAP appeared to be highly expressed in cells adjacent to the wing margin, there are concerns whether the observed regulation of EGFRAP by Notch signaling is cell autonomous and whether the model needs to be modified. Because of these concerns, we will not be able to accept this version of the manuscript, but we would be willing to review a much-revised version. We cannot, of course, promise publication at that time.

If you decide to revise the manuscript for further consideration at PLOS Genetics, please aim to resubmit within the next 60 days, unless it will take extra time to address the concerns of the reviewers, in which case we would appreciate an expected resubmission date by email to plosgenetics@plos.org.

[LINK]

We are sorry that we cannot be more positive about your manuscript at this stage. Please do not hesitate to contact us if you have any concerns or questions.

Yours sincerely,

Wei Du

Guest Editor

PLOS Genetics

Gregory P. Copenhaver

Editor-in-Chief

PLOS Genetics

Reviewer's Responses to Questions

**Comments to the Authors:**

Reviewer #1: By addressing the reviewers comments, the authors have significantly improved the paper. It is now presenting the findings from a more balanced angle that is appropriate for publication.

Regarding the point raised by Reviewer 2 about the credibility of the 2-hybrid assay for SH2/P-Tyr binding, I did not find the author's comments convincing. Instead of placing this finding as a proof for EGFRAP/EGFR association, it would be better to say after the initial identification of EGFRAP in the activated Ras screen, that previously a 2-hybrid assay has also identified an interaction between the two proteins (Supp material), and not use it as an assay for structure-function studies of the interaction between EGFRAP and EGFR.

Reviewer #2: The authors have made a good effort to address the points raised by all the reviewers, and have significantly improved the manuscript.

Reviewer #3: In this revised version, the authors have responded to the previous review. This led to additional experiments, which have quite radically changed the understanding of the biological role and control of EGFRAP. While this is progress, it is worrying that the previous version, if it had been published, would instantly have been in need of extensive revision. The new findings call for a different ‘angle’ on the results. EGFRAP appears to act redundantly with PVRAP as a negative regulator of EGFR. The redundant function of these genes therefore provides buffering and robustness in normal physiology. And that is quite different than a role that becomes evident only in an oncogenic background—so the title does not seem to fit the current version. This is a major concern.

Other points:

1) The Y-2-H experiment does not warrant a main figure and should be a supplement, as the authors note it is just a beginning step. Even if EGFRAP and EGFR would have to be overexpressed in S2 cells—this is still a superior experiment and an S2 line expressing EGFR already exists.

2) In contrast to the Y-2-H experiment, the redundancy with PVRAP is an important result and should be a main figure. This type of redundancy, in neighboring genes has many precedents and as described above explains the lack of a phenotype in either gene alone.

3) The discussion relative to human cancer still appears underdeveloped. For example, the second sentence in the abstract—' However, activated Ras alone is insufficient to produce malignancy.’ The multi-hit nature of cancer is well known and activation of Ras in the absence of other genetic alterations causes oncogene-induced senescence and indeed there may be an optimal level of Ras activation for tumorigenesis (comment on recent study in eLife: https://elifesciences.org/articles/69192 ). EGFRAP and PVRAP appear to a case of two related genes that together have tumor suppressor function.

**Have all data underlying the figures and results presented in the manuscript been provided?**

Reviewer #1: Yes

Reviewer #2: **No: **The numerical data underlying the graphs presented are not included.

Reviewer #3: Yes

PLOS authors have the option to publish the peer review history of their article (what does this mean?). If published, this will include your full peer review and any attached files.

Reviewer #1: No

Reviewer #2: No

Reviewer #3: No

---

## [Decision Letter · Decision Letter 2]

23 Jul 2021

Dear Dr Martin-Bermudo,

We are pleased to inform you that your manuscript entitled "EGFRAP encodes a new negative regulator of the EGFR acting in both normal and oncogenic EGFR/Ras-driven tissue morphogenesis" has been editorially accepted for publication in PLOS Genetics. Congratulations!

Yours sincerely,

Wei Du

Guest Editor

PLOS Genetics

Gregory P. Copenhaver

Editor-in-Chief

PLOS Genetics

Comments from the reviewers (if applicable):

Reviewer's Responses to Questions

**Comments to the Authors:**

Reviewer #1: The authors have addressed the points raised in the last revision round.

Reviewer #2: The authors have appropriately addressed all the reviewers' comments.

Reviewer #3: The authors have paid close attention to the review comments and now have a much better understanding of EGFRAP function. Extensive refinement of the original interpretation and new results have resulted in a more solid biological understanding of the gene's role in regulation.

**Have all data underlying the figures and results presented in the manuscript been provided?**

Reviewer #1: Yes

Reviewer #2: Yes

Reviewer #3: Yes

PLOS authors have the option to publish the peer review history of their article (what does this mean?). If published, this will include your full peer review and any attached files.

Reviewer #1: No

Reviewer #2: No

Reviewer #3: No

**Data Deposition**

http://datadryad.org/submit?journalID=pgenetics&manu=PGENETICS-D-21-00125R2

**Press Queries**

---

## [Editor Report · Acceptance letter]

13 Aug 2021

PGENETICS-D-21-00125R2 

EGFRAP encodes a new negative regulator of the EGFR acting in both normal and oncogenic EGFR/Ras-driven tissue morphogenesis 

Dear Dr Martin-Bermudo, 

We are pleased to inform you that your manuscript entitled "EGFRAP encodes a new negative regulator of the EGFR acting in both normal and oncogenic EGFR/Ras-driven tissue morphogenesis" has been formally accepted for publication in PLOS Genetics! Your manuscript is now with our production department and you will be notified of the publication date in due course.

With kind regards,

Andrea Szabo

PLOS Genetics

On behalf of:
